# Quantitative Convergences of Lie Group Momentum Optimizers

**Lingkai Kong**
School of Mathematics
Georgia Institute of Technology
lkong75@gatech.edu

**Molei Tao**
School of Mathematics
Georgia Institute of Technology
mtao@gatech.edu

## Abstract

Explicit, momentum-based dynamics that optimize functions defined on Lie groups can be constructed via variational optimization and momentum trivialization. Structure preserving time discretizations can then turn this dynamics into optimization algorithms. This article investigates two types of discretization, Lie Heavy-Ball, which is a known splitting scheme, and Lie NAG-SC, which is newly proposed. Their convergence rates are explicitly quantified under $L$-smoothness and *local* strong convexity assumptions. Lie NAG-SC provides acceleration over the momentumless case, i.e. Riemannian gradient descent, but Lie Heavy-Ball does not. When compared to existing accelerated optimizers for general manifolds, both Lie Heavy-Ball and Lie NAG-SC are computationally cheaper and easier to implement, thanks to their utilization of group structure. Only gradient oracle and exponential map are required, but not logarithm map or parallel transport which are computational costly.[1]

## 1 Introduction

First-order optimization, i.e., with the gradient of the potential (i.e. the objective function) given as the oracle, is ubiquitously employed in machine learning. Within this class of algorithms, momentum is often introduced to accelerate convergence; for example, in Euclidean setups, it has been proved to yield the optimal dimension-independent convergence order, for a large class of first-order optimizers, under strongly convex and $L$-smooth assumptions of the objective function[21, Sec. 2].

Gradient Descent (GD) without momentum can be generalized to Riemannian manifold by moving to the negative gradient direction using the exponential map with a small step size. This generalization is algorithmically straight forward, but quantifying the convergence rate in curved spaces needs nontrivial efforts [6, 32, 27]. In comparison, generalizing momentum GD to manifold itself is nontrivial due to curved geometry; for example, the iteration must be kept on the manifold and the momentum must stay in the tangent space, which changes with the iteration, at the same time. It is even more challenging to quantify the convergence rate theoretically due to the loss of linearity in the space, leading to the lack of triangle inequality and cosine rule. Finally, there is not necessarily acceleration unless the generalization is done delicately.

Regardless, optimization on manifolds is an important task, for which the manifold structure can either naturally come from the problem setup or be artificially introduced. A simple but extremely important example is to compute the leading eigenvalues of a large matrix, which can be approached efficiently via optimization on the Stiefel manifold [19]; a smaller scale version can also be solved via optimization on $\mathsf{SO}(n)$ [7, 29, 8, 26]. More on the modern machine learning side, one can algorithmically add orthonormal constraints to deep learning models to improve their accuracy and robustness [e.g., 5, 9, 17, 19, 24]. Both examples involve $\mathsf{SO}(n)$, which is an instance of an important type of curved spaces called Lie groups.

---

[1] Code can be found at `https://github.com/konglk1203/Accelerated_Optimizer_On_Lie_Group`

38th Conference on Neural Information Processing Systems (NeurIPS 2024).

Lie groups are manifolds with additional group structure, and the nonlinearity of the space manifests through the non-commutative group multiplication. The group structure can help not only design momentum optimizer but also analyze its convergence. More precisely, this work will make the following contributions:

- Provide the first quantitative analysis of Lie group momentum optimizers. This is significant, partly because there is no nontrivial convex functions on many Lie groups (see Rmk. 1), so we have to analyze nonconvex optimization.

- Theoretically show an intuitively constructed momentum optimizer, namely Lie Heavy-Ball, may not yield accelerated convergence. Numerical evidence is also provided (Sec. 6).

- Generalize techniques from Euclidean optimization to propose a Lie group optimizer, Lie NAG-SC, that provably has acceleration.

Comparing to other optimizers that are designed for general manifolds, we bypass the requirements for costly operations such as parallel transport [e.g., 4], which is a way to move the momentum between tangent spaces, and the computation of geodesic [e.g., 1], which may be an issue due to not only high computational cost but also its possible non-uniqueness.

## 1.1 Related work

In Euclidean space, Gradient Descent (GD) for $\mu$-strongly convex and $L$-smooth objective function can converge as $\|x_k - x_*\| \le \left(1 - C\frac{\mu}{L}\right)^k \|x_k - x_0\|$ [2] upon appropriately chosen learning rate. Momentum can accelerate the convergence by softening the dependence on the condition number $\kappa := L/\mu$. However, how momentum is introduced matters to achieving such an acceleration. For example, NAG-SC [21] has convergence rate [3] $1 - C\sqrt{\frac{\mu}{L}}$, but Heavy-Ball [23] still has linear dependence on the conditional number, similar to gradient descent without momentum (but it may work better for nonconvex cases).

Remarkable quantitative results also existed for manifold optimization. The momentum-less case is relatively simpler, and [32], for example, developed convergence theory for GD on Riemannian manifold under various assumptions on convexity and smoothness, which matched the classical Euclidean result — for instance, Thm. 15 of their paper gave a convergence rate of $1 - C \min\left\{\frac{\mu}{L}, K\right\}$ [4] when the learning rate is $1/L$, under geodesic-$\mu$-strong-convexity and geodesic-$L$-smoothness. For the momentum case, [3] analyzed the convergence of a related dynamics in continuous time, namely an optimization ODE corresponding to momentum gradient flow, on Riemannian manifolds under both geodesically strongly and weakly convex potentials, based on a tool of modified cosine rule. However, numerical methods in discrete time that are easy and cheap to implement and provably convergent in an accelerated fashion under mild conditions are still under-developed. One existing idea is to transform a function on the manifold to a function on a Euclidean space by the logarithm function. More precisely, in the case where the logarithm is a one-to-one map from the manifold to the tangent space, it can be used to project the objective function on the manifold to a function on the tangent space ('pullback objective function'), enabling the usage of accelerated algorithms in Euclidean spaces [e.g., 11]. Although such analysis may relax the requirement of global convexity, it requires assumptions on the 'pullback objective function', which is hard to check in reality. Another series of seminal works include [33, 1], which analyzed the convergence of a class of optimizers by extending Nesterov's technique of estimating sequence [21] to Riemannian manifolds. They managed to show a convergence rate between $1 - C\sqrt{\frac{\mu}{L}}$ and $1 - C\frac{\mu}{L}$, i.e., with conditional number dependence inbetween that of GD with and without momentum in the Euclidean cases. They further proved that, as the iterate gets closer to the minimum, the rate bound gets better because it converges to $1 - C\sqrt{\frac{\mu}{L}}$. However, their algorithm requires the logarithm map (inverse of the exponential map), which may not be uniquely defined on many manifolds (e.g., sphere) and can be computationally expensive. In contrast, Lie NAG-SC, which we will construct, works only for Lie group manifolds, but they are

---

[2]All the constants $C$ in this paper may be different case-by-case, but they are all independent of dimension and the condition number.

[3]For continuous dynamics, convergence rate means $c$ if $U(g_t) - U(g_*) = \mathcal{O}(e^{-ct})$. For discrete algorithms, it refers to $c$ if $U(g_k) - U(g_*) = \mathcal{O}(c^k)$.

[4]We will use $K$ to denote constants that depend on the manifold structure, e.g., diameter, curvature. It may be different case-by-case.

more efficient when applicable, due to being based on only exponential map and gradient oracle. Acceleration of the same type will also be theoretically proved.

Momentum optimizers specializing in Lie groups have also been constructed before [26], where variational optimization [30] was generalized to the manifold setup and then left trivialization were employed to obtain ODEs with Euclidean momentum that perform optimization in continuous time. Our work is also based on those ODEs, whose time however has to be discretized so that an optimization algorithm can be constructed. Delicate discretizations have been proposed in [26] so that the optimization iterates stay exactly on the manifold, saving computational cost and reducing approximation errors. But we will further improve those discretizations. More precisely, note first that [26] slightly abused notation and called both the continuous dynamics and one discretization Lie NAG-SC. However, we find that their splitting discretization may not give the best optimizer – at least, a 1st-order version of their splitting scheme yields a linear condition number dependence in our convergence rate bound. Since this splitting-based optimizer almost degenerates to heavy-ball in the special case of Euclidean spaces (as Rmk. 28 will show), we refine the terminology and call it Lie Heavy-Ball. To remedy the lack of acceleration, we propose a new discretization that actually has square root condition number dependence and thus provable acceleration, and call it as (the true) Lie NAG-SC.

Finally, note there can be obstructions to accelerated optimization, for example roughly when curvature is negative [16, 10, 12]. That is, however, not a contradiction because our setup involves positive curvature.

## 1.2 Main results

We consider the local minimization of a differentiable function $U : \mathsf{G} \to \mathbb{R}$, i.e., $\min_{g \in \mathsf{G}} U(g)$, where $\mathsf{G}$ is a finite-dimensional compact Lie group, and the oracle allowed is the differential of $U$.

Two optimizers we focus on are given in Alg. (1). Under assumptions of $L$-smoothness and locally geodesic-$\mu$-strong convexity, we proved that Lie Heavy-Ball has convergence rate $\left(1 + C\frac{L}{\mu}\right)^{-1}$, which is approximately the convergence rate of $1 - C\min\left\{\frac{L}{\mu}, K\right\}$ for Lie GD (Eq.1, which is identical to Riemannian GD applied to Lie groups); this is no acceleration. To accelerate, we propose a new Lie NAG-SC algorithm, with provable convergence rate $\left(1 + C\min\left\{\sqrt{\frac{L}{\mu}}, K\right\}\right)^{-1}$. Note the condition number dependence becomes $\sqrt{\kappa}$ instead of $\kappa := L/\mu$, hence acceleration.

For a summary of our main results, please see Table 1.

---

**Algorithm 1:** Momentum optimizer on Lie groups

---

**Parameter** : step size $h > 0$, friction $\gamma > 0$, number of iterations $N$
**Initialization** : $g_0 \in \mathsf{G}$, $\xi_0 = 0$
**Output** : Local minimum of $U$
**for** $k = 0, \cdots, N - 1$ **do**
    **if** *Heavy-Ball* **then**
        $\xi_{k+1} = (1 - \gamma h)\xi_k - h T_{g_k} \mathsf{L}_{g_k^{-1}} \nabla U(g_k)$
    **if** *NAG-SC* **then**
        $\xi_{k+1} = (1 - \gamma h)\xi_k - (1 - \gamma h)h\left(T_{g_k} \mathsf{L}_{g_k^{-1}} \nabla U(g_k) - T_{g_{k-1}} \mathsf{L}_{g_{k-1}^{-1}} \nabla U(g_{k-1})\right) -$
        $h T_{g_k} \mathsf{L}_{g_k^{-1}} \nabla U(g_k)$
    $g_{k+1} = g_k \exp(h\xi_{k+1})$
**end**
**return** $g_N$

---

**Remark 1** (Triviality of convex functions on Lie groups). *We do not assume any global convexity of $U$. In fact, $U$ has to be nonconvex for any meaningful optimization to happen. This is because we are considering compact Lie groups[5], and a convex function on a connected compact manifold could only be a constant function [31]. An intuition for this is, a convex function on a closed geodesic must be constant. See [e.g., 18, Sec. B.3] for more discussions. Our analysis is, importantly, for nonconvex $U$, and convexity is only required locally to ensure a quantitative rate estimate can be obtained.*

---

[5]Even if we consider noncompact Lie groups, many of them have compact Lie subgroups [20], and our argument would still hold.

| | Continuous dynamics | Heavy-Ball | NAG-SC |
|---|---|---|---|
| Scheme | Eq. (2) | Eq. (9) | Eq. (13) |
| Step size $h$ | - | $\frac{\sqrt{\mu}}{4L}$ | $\min\left\{\frac{1}{\sqrt{2L}}, \frac{1}{2p(a)}\right\}$ |
| Convergence rate $c$ | $\frac{2}{3}\sqrt{\mu}$ | $\left(1+\frac{\mu}{16L}\right)^{-1}$ | $\left(1+\frac{1}{30}\sqrt{\mu}\min\left\{\frac{1}{\sqrt{2L}}, \frac{1}{2p(a)}\right\}\right)^{-1}$ |
| (Modified) energy | Eq. (5) | Eq. (10) | Eq. (41) [6] |
| Lyapunov function | Eq. (6) | Eq. (12) | Eq. (14) |
| Main theorem | Thm. 9 | Thm. 13 | Thm. 14 |

Table 1: A summary of main results

## 2 Preliminaries and setup

### 2.1 Lie group and Lie algebra

A **Lie group**, denoted by $\mathsf{G}$, is a differentiable manifold with a group structure. A **Lie algebra** is a vector space with a bilinear, alternating binary operation that satisfies the Jacobi identity, known as Lie bracket. The tangent space at $e$ (the identity element of the group) is a **Lie algebra**, denoted as $\mathfrak{g} := T_e\mathsf{G}$. The dimension of the Lie group $\mathsf{G}$ will be denoted by $m$.

**Assumption 2** (general geometry). *We assume the Lie group $\mathsf{G}$ is finite-dimensional and compact.*

One technique we will use to handle momentum is called **left-trivialization**: Left group multiplication $L_g : \hat{g} \to g\hat{g}$ is a smooth map from the Lie group to itself and its tangent map $T_{\hat{g}}L_g : T_{\hat{g}}\mathsf{G} \to T_{g\hat{g}}\mathsf{G}$ is a one-to-one map. As a result, for any $g \in \mathsf{G}$, we can represent the vectors in $T_g\mathsf{G}$ by $T_eL_\xi$ for $\xi \in T_e\mathsf{G}$. This operation is the left-trivialization. It comes from the group structure and may not exist for a general manifold. If the group is represented via an embedding to matrix group, i.e., $g, \xi \in \mathbb{R}^{n \times n}$, then the left trivialization is simply given by $T_eL_g\xi = g\xi$ with the right-hand side given by matrix multiplication.

A Riemannian metric is required to take Riemannian gradient and we are considering a left-invariant metric: we first define an inner product $\langle \cdot, \cdot \rangle$ on $\mathfrak{g}$, which is a linear space, and then move it around by the differential of left multiplication, i.e., the inner product at $T_g\mathsf{G}$ is for $\eta_1, \eta_2 \in T_g\mathsf{G}$, $\langle \eta_1, \eta_2 \rangle := \langle T_gL_{g^{-1}}\eta_1, T_gL_{g^{-1}}\eta_2 \rangle$.

### 2.2 Optimization dynamics

Riemannian GD [e.g., 32] with iteration $g_{k+1} = \exp_{g_k}(-h\nabla U(g_k))$ can be employed to optimize $U$ defined on $\mathsf{G}$, where $\nabla$ is Riemannian gradient, and $\exp : \mathsf{G} \times T_g\mathsf{G} \to \mathsf{G}$ is the exponential map. To see a connection to the common Euclidean GD, it means we start from $g_k$ and go to the direction of negative gradient with step size $h$ to get $g_{k+1}$ by geodesic instead of straight line. Riemannian GD can be understood as a time discretization of the Riemannian gradient flow dynamics $\dot{g} = -\nabla U(g)$.

In the Lie group case, it is identical to the following Lie GD obtained from left-trivialization [26]:

$$g_{k+1} = g_k \exp_e(hT_{g_k}L_{g_k^{-1}}\nabla U(g_k)), \tag{1}$$

where $T_{g_k}L_{g_k^{-1}}\nabla U(g_k) \in \mathfrak{g}$ is the left-trivialized gradient. $\exp_e$[7] is the exponential map staring at the group identity $e$ following the Riemannian structure given by the left-invariant metric, and the operation between $g_k$ and $\exp(hT_{g_k}L_{g_k^{-1}}\nabla U(g_k))$ is the group multiplication. To accelerate its convergence, momentum was introduced to the Riemannian gradient flow via variational optimization and left-trivialization [26], leading to the following dynamics:

$$\begin{cases} \dot{g} = T_eL_g\xi \\ \dot{\xi} = -\gamma(t)\xi + \mathrm{ad}_\xi^* \xi - T_gL_{g^{-1}}\nabla U(g) \end{cases} \tag{2}$$

Here $g(t) \in \mathsf{G}$ is the position variable. $\dot{g}$ is the standard 'momentum' variable even though it should really be called velocity. It lives $T_{g(t)}\mathsf{G}$, which varies as $g(t)$ changes in time, and we will utilize group structure to avoid this complication. More precisely, the dynamics lets the 'momentum' $\dot{g}$ be $T_eL_g\xi$, and $\xi$ is therefore $T_gL_{g^{-1}}\dot{g}$ and it is our new, left-trivialized momentum. Intuitively, one can

---

[6]The monotonicity of this energy function requires smaller step size than it listed in this table. See discussion in Rmk. 36 and the details are provided in Sec. D.1

[7]The group exponential map and the exponential map from Riemannian structure can be different [18, Sec. D.1]. However, under our choice of the left-invariant metric later in Lemma 3, they are identical and $\exp$ will be the group exponential unless further specified.

think $\xi$ as angular momentum, and $T_e L_g \xi$ being $g\xi$ is position times angular momentum, which is momentum. Similar to the Lie GD Eq. (1), we will not use $\nabla U(g)$ directly, but its left-trivialization $T_g L_{g^{-1}}(\nabla U(g))$, to update the left-trivialized momentum.

This dynamics essentially models a damped mechanical system, and Tao and Ohsawa [26] proved this ODE converges to a local minimum of $U$ using the fact that the total energy (kinetic energy $\frac{1}{2}\langle \xi, \xi \rangle$ plus potential energy $U$) is drained by the friction term $-\gamma\xi$. In general, $\gamma$ can be a positive time-dependent function (e.g., for optimizing convex but not strongly-convex functions), but for simplicity, we will only consider locally strong-convex potentials, and constant $\gamma$ is enough.

For curved space, an additional term $\mathrm{ad}_\xi^* \xi$ that vanishes in Euclidean space shows up in Eq. (2). It could be understood as a generalization of Coriolis force that accounts for curved geometry and is needed for free motion. The **adjoint operator** $\mathrm{ad} : \mathfrak{g} \times \mathfrak{g} \to \mathfrak{g}$ is defined by $\mathrm{ad}_X Y := [X, Y]$. Its dual, known as the **coadjoint operator** $\mathrm{ad}^* : \mathfrak{g} \times \mathfrak{g} \to \mathfrak{g}$, is given by $\langle \mathrm{ad}_X^* Y, Z \rangle = \langle Y, \mathrm{ad}_X Z \rangle, \forall Z \in \mathfrak{g}$.

### 2.3 Property of Lie groups with $\mathrm{ad}^*$ skew-adjoint

The term $\mathrm{ad}_\xi^* \xi$ in the optimization ODE (2) is a quadratic term and it will make the numerical discretization that will be considered later difficult. Another complication from this term is, it depends on the Riemannian metric, and indicates an inconsistency between the Riemannian structure and the group structure, i.e., the exponential map from the Riemannian structure is different from the exponential map from the group structure. Fortunately, on a compact Lie group, the following lemma shows a special metric on $\mathfrak{g}$ can be chosen to make the term $\mathrm{ad}_\xi^* \xi$ vanish.

**Lemma 3** (ad skew-adjoint [20]). *Under Assumption 2, there exists an inner product on $\mathfrak{g}$ such that the operator* $\mathrm{ad}$ *is skew-adjoint, i.e.,* $\mathrm{ad}_\xi^* = -\mathrm{ad}_\xi$ *for any $\xi \in \mathfrak{g}$.*

This special inner product will also give other properties useful in our technical proofs; see Sec. A.1.

### 2.4 Assumption on potential function

To show convergence and quantify its rate for the discrete algorithm, some smoothness assumption is needed. We define the $L$-smoothness on a Lie group as the following.

**Definition 4** ($L$-smoothness). *A function $U : \mathsf{G} \to \mathbb{R}$ is $L$-smooth if and only if $\forall g, \hat{g} \in G$,*

$$\left\| T_{\hat{g}} L_{\hat{g}^{-1}} \nabla U(\hat{g}) - T_g L_{g^{-1}} \nabla U(g) \right\| \le L d(\hat{g}, g) \tag{3}$$

*where $d$ is the geodesic distance.*

Under the choice of metric in Lemma 3 that $\mathrm{ad}$ is skew-adjoint, Lemma 21 shows this is same as the commonly used geodesic-$L$-smoothness (Def. 20).

To provide an explicit convergence rate, some convex assumption on the objective function is usually needed. Under the assumption of unique geodesic on a geodesically convex set $S \subset \mathsf{G}$, the definition of strongly convex functions in Euclidean spaces can be generalized to Lie groups:

**Definition 5** (Locally geodesically strong convexity). *A function $U : \mathsf{G} \to \mathbb{R}$ is locally geodesic-$\mu$-strongly convex at $g_*$ if and only if there exists a geodesically convex neighbourhood of $g_*$, denoted by $S$, such that $\forall g, \hat{g} \in S$,*

$$U(g) - U(\hat{g}) \ge \left\langle T_{\hat{g}} L_{\hat{g}^{-1}} \nabla U(\hat{g}), \log \hat{g}^{-1} g \right\rangle + \frac{\mu}{2} \left\| \log \hat{g}^{-1} g \right\|^2 \tag{4}$$

*where $\log$ is well-defined due to the geodesic convexity of $S$.*

## 3 Convergence of the optimization ODE in continuous time

To start, we provide a convergence analysis of the ODE (2), since our numerical scheme comes from its time discretization. We do not claim such convergence analysis for the ODE is new, and in fact, convergence for continuous dynamics has been provided on general manifolds [e.g., 3]. However, we will prove it using our technique to be self-contained and provide some insights for the convergence analysis of the discrete algorithm later.

Define the total energy $E^{\mathrm{ODE}} : \mathsf{G} \times \mathfrak{g} \to \mathbb{R}$ as

$$E^{\mathrm{ODE}}(g, \xi) := U(g) + \frac{1}{2} \|\xi\|^2 \tag{5}$$

i.e., the total energy is the sum of the potential energy and the kinetic energy. Thanks to the friction $\gamma$, the total energy is monotonely decreasing, which provides global convergence to a stationary point.

**Theorem 6** (Monotonely decreasing of total energy [26]). *Suppose the potential function $U \in \mathcal{C}^1(\mathsf{G})$ and the trajectory $(g(t), \xi(t))$ follows ODE (2). Then*

$$\frac{d}{dt} E^{ODE}(g(t), \xi(t)) = -\gamma \|\xi\|^2$$

Thm. 6 provides the global convergence of ODE (2) to a stationary point under only $\mathcal{C}^1$ smoothness: when the system converges, we have $\xi_\infty = 0$, which gives $\|\nabla U(g_\infty)\| = 0$.

Moreover, using the non-increasing property of total energy, the following corollary states that if the particle starts with small initial energy, it will be trapped in a sub-level set of $U$. The local potential well can be defined using $U$'s sub-level set.

**Definition 7** ($u$ sub-level set). *Given $u \in \mathbb{R}$, we define the $u$ sub-level set of $U$ as*

$$\{g \in \mathsf{G} : U(g) \le u\} := \bigcup_{i \ge 0} S_i$$

*i.e. a disjoint union of connected components.*

**Corollary 8.** *Suppose $U \in \mathcal{C}^1(\mathsf{G})$. Let $u = E^{ODE}(g(0), \xi(0))$. If the $u$ sub-level set of $U$ is $\bigcup_{i \ge 0} S_i$ and $g(0) \in S_0$, then we have $g(t) \in S_0, \forall t \ge 0$.*

Under further assumption of local strong convexity on this sub-level set, convergence rate can be quantified via a Lyapunov analysis inspired by [25]. More specifically, given a fixed local minimum $g_*$, there is provably a local unique geodesic convex neighbourhood of $g_*$. Denote it by $S$, and we define $\mathcal{L}^{\text{ODE}}$ on $S$ by

$$\mathcal{L}^{\text{ODE}}(g, \xi) := U(g) - U(g_*) + \frac{1}{4}\|\xi\|^2 + \frac{1}{4}\left\|\gamma \log g_*^{-1} g + \xi\right\|^2 \tag{6}$$

By assuming the local geodesic-$\mu$-strong convexity of $U$ on $S$, we have the following quantification of Eq. (2).

**Theorem 9** (Convergence rate of the optimization ODE). *If the initial condition $(g_0, \xi_0)$ satisfies that $g_0 \in S$ for some geodesically convex set $S \subset \mathsf{G}$, $U \in \mathcal{C}^1(\mathsf{G})$ is locally geodesic-$\mu$-convex on $S$, and the $u$ sub-level set of $U$ with $u = E^{ODE}(g_0, \xi_0)$ satisfies $S_0 \subset S$, then we have*

$$U(g(t)) - U(g_*) \le e^{-c_{ODE}t}\mathcal{L}^{ODE}(g_0, \xi_0) \tag{7}$$

*with $c_{ODE} = \frac{2}{3}\sqrt{\mu}$ by choosing $\gamma = 2\sqrt{\mu}$.*

**Remark 10.** *This theorem alone is a local convergence result and a {Lie group + momentum} extension of an intuitive result for Euclidean gradient flow, which is, if the initial condition is close enough to a minimizer and the objective function has a positive definite Hessian at that minimizer, then gradient flow converges exponentially fast to that minimizer. However, Thm.6 already ensures global convergence, and if not stuck at a saddle point, the dynamics will eventually enter some local potential well. If that potential well is locally strongly convex at its minimizer, then the local convergence result (Thm.9) supersedes the global convergence result (which has no rate), and gives the asymptotic convergence rate. Note however that different initial conditions may lead to convergence to different potential wells (and hence minimizers), as usual.*

## 4 Convergence of Lie Heavy-Ball/splitting discretization in discrete time

One way to obtain a manifold optimization algorithm by time discretization of the ODE (2) is to split its vector field as the sum of two, and use them respectively to generate two ODEs:

$$\begin{cases} \dot{g} = T_e \mathsf{L}_g \xi \\ \dot{\xi} = 0 \end{cases} \quad \begin{cases} \dot{g} = 0 \\ \dot{\xi} = -\gamma \xi - T_g \mathsf{L}_{g^{-1}} \nabla U(g) \end{cases} \tag{8}$$

Each ODE enjoys the feature that its solution stays exactly on $\mathsf{G} \times \mathfrak{g}$ [26], and therefore if one alternatively evolves them for time $h$, the result is a step-$h$ time discretization that exactly respects the geometry (no projection needed). If one approximates $\exp(-\gamma h)$ by $1 - h\gamma$, then the same property holds, and the resulting optimizer is

$$\begin{cases} g_{k+1} = g_k \exp(h\xi_{k+1}) \\ \xi_{k+1} = (1 - \gamma h)\xi_k - hT_{g_k}\mathsf{L}_{g_k^{-1}}\nabla U(g_k) \end{cases} \tag{9}$$

In Euclidean cases, such numerical scheme can be viewed as Polyak's Heavy-Ball algorithm after a change of variable (Rmk. 27), and will thus be referred to as Lie Heavy-Ball. It is also a 1st-order (in $h$) version of the '2nd-order Lie-NAG' optimizer in [26] (Rmk. 28).

To analyze Lie Heavy-Ball's convergence, we again seek some 'energy' function such that the iteration of the numerical scheme Eq. (9) will never escape a sub-level set of the potential, similar to the continuous case. Given fixed friction parameter $\gamma$ and step size $h$, we define the modified energy $E^{HB} : \mathsf{G} \times \mathfrak{g} \to \mathbb{R}$ as

$$E^{HB}(g, \xi) := U(g) + \frac{(1 - \gamma h)^2}{2} \|\xi\|^2 \tag{10}$$

**Theorem 11** (Monotonely decreasing of modified energy of Heavy Ball)**.** *Assume the potential $U$ is globally $L$-smooth. When the step size satisfies $h \le \frac{\gamma}{\gamma^2 + L}$, we have the modified energy $E^{HB}$ is monotonely decreasing, i.e.,*

$$E^{HB}(g_k, \xi_k) - E^{HB}(g_{k-1}, \xi_{k-1}) \le -\gamma h \|\xi_k\|^2$$

Thm. 11 provides the global convergence of Heavy-Ball scheme Eq. (9) to a stationary point under only $L$-smoothness: Due to the monotonicity of the energy function $E^{HB}$, the system will eventually converge. When it converges, since $g$ is not moving, we have $\|\xi_\infty\| = 0$, leading to the fact that $\|\nabla U(g_\infty)\| = 0$. More importantly, the following corollary shows that the non-increasing property of the modified traps $g$ in sub-level set of $U$:

**Corollary 12.** *Let $u = E^{HB}(g_0, \xi_0)$. If the $u$ sub-level set of $U$ satisfies $g_0 \in S_0$ and*

$$d\left(S_0, \bigcup_{i \ge 1} S_i\right) > h\sqrt{2E^{HB}(g_0, \xi_0)} + h^2 \max_{S_0} \|\nabla U\| \tag{11}$$

*Then we have $g_k \in S_0$ for any $k$ for the Heavy-Ball scheme Eq. (9) when $h \le \frac{\gamma}{\gamma^2 + L}$.*

Under the further assumption of local strong convexity on this sub-level set, the convergence rate can be quantified via a Lyapunov analysis inspired by [25]. More specifically, given a fixed local minimum $g_*$, there is a local unique geodesic neighbourhood of $g_*$, denoted by $S$, and we define $\mathcal{L}^{HB}$ on $S$ by

$$\mathcal{L}^{HB}(g, \xi) := \frac{1}{1 - \gamma h}\left(U(g\exp(-h\xi)) - U(g_*)\right) + \frac{1}{4}\|\xi\|^2 + \frac{1}{4}\left\|\frac{\gamma}{1 - \gamma h}\log g_*^{-1}g + \xi\right\|^2 \tag{12}$$

The exponential decay for the Lyapunov function (Lemma 32) helps us quantify of the convergence rate for Eq. (9) in the following theorem:

**Theorem 13** (Convergence rate of Heavy-Ball scheme)**.** *If the initial condition $(g_0, \xi_0)$ satisfies that $g_0 \in S$ for some geodesically convex set $S \subset \mathsf{G}$, $U$ is $L$-smooth and locally geodesic-$\mu$-convex on $S$, and the $u$ sub-level set of $U$ with $u = E^{ODE}(g_0, \xi_0)$ satisfies $S_0 \subset S$ and Eq. (11), then we have*

$$U(g_k) - U(g_*) \le c_{HB}^k \mathcal{L}^{HB}(g_0, \xi_0)$$

*with $c_{HB} := \left(1 + \frac{\mu}{16L}\right)^{-1}$ by choosing $\gamma = 2\sqrt{\mu}$, $h = \frac{\sqrt{\mu}}{4L}$.*

Note the rate is $(1 + 1/(16\kappa))^{-1}$. The condition number dependence is linear ($\kappa$) but not $\sqrt{\kappa}$. Similarly, the procedure of global convergence → local potential well → local minimum discussed in Rmk. 10 also applies the Heavy-Ball algorithm.

## 5 Convergence of Lie NAG-SC in discrete time

The motivation for NAG-SC is to improve the condition number dependence. The convergence rate of Heavy-Ball shown in Thm. 13 is the same as the momentumless case [e.g., 32, Thm. 15] under the assumption of local strong convexity and $L$-smoothness. To improve the condition number dependence, inspired by [25], we define Lie NAG-SC as the following:

$$\begin{cases} g_{k+1} = g_k \exp(h\xi_{k+1}) \\ \xi_{k+1} = (1 - \gamma h)\xi_k - (1 - \gamma h)h\left(T_{g_k}\mathsf{L}_{g_k^{-1}}\nabla U(g_k) - T_{g_{k-1}}\mathsf{L}_{g_{k-1}^{-1}}\nabla U(g_{k-1})\right) - hT_{g_k}\mathsf{L}_{g_k^{-1}}\nabla U(g_k) \end{cases} \tag{13}$$

Comparing to Lie Heavy-Ball, an extra $\mathcal{O}(h^2)$ term $h\left(T_{g_k}\mathsf{L}_{g_k^{-1}}\nabla U(g_k) - T_{g_{k-1}}\mathsf{L}_{g_{k-1}^{-1}}\nabla U(g_{k-1})\right)$ is introduced (see [25, Sec. 2] for more details in the Euclidean space). Our technique of left-trivialized (and hence Euclidean) momentum allows this trick to transfer directly from Euclidean to the Lie group case.

For NAG-SC, we will only provide a local convergence with quantified convergence under $L$-smoothness and local geodesically convexity on a geodesically convex subset $S \subset \mathsf{G}$. The difficulty in designing a modified energy and proving the global convergence will be given later in Rmk. 36. We define the following Lyapunov function:

$$\mathcal{L}^{\text{NAG-SC}}(g,\xi) := \frac{1}{1-\gamma h}\left(U(g\exp(-h\xi)) - U(g_*)\right) + \frac{1}{4}\|\xi\|^2 \tag{14}$$

$$+ \frac{1}{4}\left\|\xi + \frac{\gamma}{1-\gamma h}\log g_*^{-1}g + h\nabla U(g\exp(-h\xi))\right\|^2 - \frac{h^2(2-\gamma h)}{4(1-\gamma h)}\|\nabla U(g\exp(-h\xi))\|^2$$

where $g_*$ is the minimum of $U$ in $S$. This Lyapunov function helps us to trap $g$ in a local potential well and quantify the convergence rate:

**Theorem 14** (Convergence rate of NAG-SC). *If the initial condition $(g_0, \xi_0)$ satisfies that $g_0 \in S$ for some geodesically convex set $S \subset \mathsf{G}$ satisfying $\max_{g\in S} d(g_*, g) \le \frac{a}{A}$ for some $a < 2\pi$ and $A := \max_{\|X\|=1}\|\text{ad}_X\|_{op}$, $U$ is $L$-smooth and locally geodesic-$\mu$-convex on $S$, and the $u$ sub-level set of $U$ with $u = (1-\gamma h)^{-1}\mathcal{L}^{\text{NAG-SC}}(g_0, \xi_0)$ satisfies $S_0 \subset S$ and*

$$d(S_0, S - S_0) > h\sqrt{\mathcal{L}^{NAG\text{-}SC}(g_0, \xi_0)} \tag{15}$$

*then we have*

$$U(g_k) - U(g_*) \le c_{NAG\text{-}SC}^k \mathcal{L}^{NAG\text{-}SC}(g_0, \xi_0)$$

*by choosing $h = \min\left\{\frac{1}{\sqrt{2L}}, \frac{1}{2p(a)}\right\}$ and $\gamma = 2\sqrt{\mu}$, with $c_{NAG\text{-}SC} := \left(1 + \frac{1}{30}\sqrt{\mu}\min\left\{\frac{1}{\sqrt{2L}}, \frac{1}{2p(a)}\right\}\right)^{-1}$, where*

$$p(x) := \frac{x}{1-\exp(-x)} \tag{16}$$

Unlike sampling ODE and Lie Heavy-Ball, monotonely decreasing modified energy is not provided for Lie NAG-SC. It is unclear whether such modified energy for NAG-SC exists, and an intuition is provided in the Rmk. 36.

Another fact in Thm. 14 that is worth noticing is, we have a term $1/p(a)$ that depends on the curvature of the Lie group [8], while the Lie Heavy-Ball has the same convergence rate as the Euclidean case [25]. It is unclear if the lost of convergence rate in Lie NAG-SC comparing to the Euclidean case is because of our proof technique or the curved space itself. However, we try to provide some insights in Rmk. 35.

# 6 Systematic numerical verification via the eigen decomposition problem

## 6.1 Analytical estimation of property of eigenvalue decomposition potential

Given a symmetric matrix, its eigen decomposition problem can be approached via an optimization problem on $\mathsf{SO}(n)$:

$$\min_{X\in\mathbb{R}^{n\times n}, X^\top X=I} \text{tr}\, X^\top B X N$$

where $N := \text{diag}([1,\ldots,n])$. This problem is a hard non-convex problem on manifold, but some analytical estimation [e.g., 7, Thm. 4] can be helpful for us to choose optimizer hyperparameters (we don't have to have those to apply the optimizers, but in this section we'd like to verify our theoretical bounds and hence $\mu$ and $L$ are needed).

This problem is non-convex with $2^n n!$ stationary points corresponding to the elements in $n$-order symmetric group, including $2^n$ local minima and $2^n$ local maxima. We suppose $B = R\Lambda R^\top$ with $\Lambda = \text{diag}\left(0, 1, \ldots, n-2, \frac{\kappa}{n-1}\right)$, where $\lambda_i$'s (the diagonal values of $\Lambda$) are in ascend order. Given $\pi$ in the $n$-symmetric group, the corresponding local minimum is $X_\pi := (X_{\pi(i)})$, i.e., we switch the columns of $X$ by $\pi$. The eigenvalues of its Hessian at the local minimum $\pi$ can be written as

$$\sigma_{ij} = (j-i)(\lambda_{\pi(j)} - \lambda_{\pi(i)}), \quad 1 \le i < j \le n$$

The global minimum is given by $\pi_* = id$ with minimum value $\sum_{i=1}^n i\lambda_i$.

---

[8] In comparison, Euclidean NAG-SC has convergence rate $\left(1 + C\sqrt{\frac{\mu}{L}}\right)^{-1}$ [25].

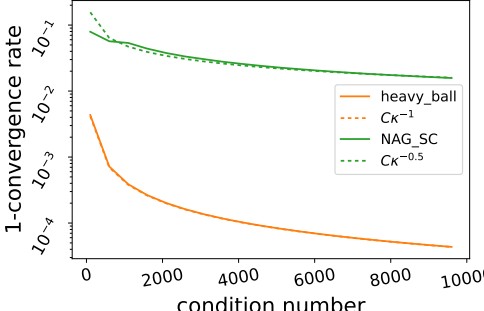 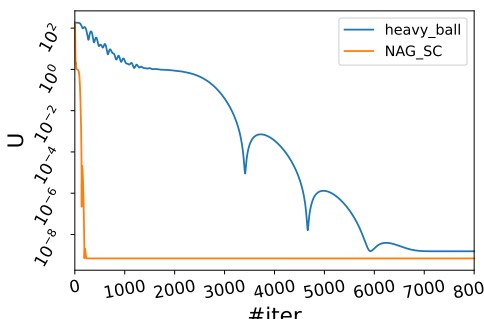

(a) Numerical estimation for $1 - c$ under different condition numbers $\kappa := \frac{L}{\mu}$ for Heavy-Ball and NAG-SC, initialized close to the global minimum. The dashed curves are fitted value using our theoretical result, i.e., for Heavy-Ball, it is fitted by $1 - c_{\text{HB}} \approx C\kappa^{-1}$ for some $C$ fit by linear regression, and for NAG-SC, it is $1 - c_{\text{NAG-SC}} \approx C\sqrt{\kappa}^{-1}$.

(b) Global convergence on non-convex potential. The initial condition is chosen closed to the global maximum, and we plot the value of the potential function along the trajectory. The horizontal tail means the algorithm converges to the machine precision.

Figure 1: Fig. 1(a) shows that 1) Lie NAG-SC converges much faster than Lie Heavy-Ball on ill-conditioned problems; 2) The fitted dashed curve and the experimental results align well, showing our theoretical analysis of the convergence rate $c_{\text{HB}}$ and $c_{\text{NAG-SC}}$ is correct. Fig. 1(b) shows the performance of our algorithms on non-convex problems experimentally. In this specific experiment, Lie NAG-SC outperforms Lie Heavy-Ball and finds the global minimum successfully without being trapped in local minimums. However, we are not sure which is better in general optimization. One possible reason for the good performance on NAG-SC is it uses a larger learning rate and is better for jumping out of the local minimums. The values of Lyapunov function along the trajectory are not provided since it is not globally defined.

## 6.2 Numerical Experiment

We use the eigenvalues at the global minimum to estimate the $L$ and $\mu$ in its neighborhood. As a result, around the global minimum, $L \approx (n-1)(\lambda_n - \lambda_1)$, and $\mu \approx \min_i\{\lambda_{i+1} - \lambda_i\}$, where we assume $\lambda$'s are sorted in the ascend order. Such estimation is used to choose our parameters ($\gamma$ and $h$) in all experiments as stated in Table 1.

Given a conditional number $\kappa := \frac{L}{\mu}$, we design $A$ in the following way: we choose $\Lambda = \text{diag}\left(0, 1, \ldots, n-2, \frac{\kappa}{n-1}\right)$ and $R$ is uniformly sampled from $\text{SO}(n)$ using [22, Sec. 2.1.1]. When the given $\kappa$ satisfies $\kappa \geq (n-1)(n-2)$, the condition number at global minimum is the given $\kappa$.

The results are presented in Fig. 1 and 2. In all experiments, we set $n = 10$, and the computations are done on a MacBook Pro (M1 chip, 8GB memory).

## 7 Application to Vision Transformer

This section will demonstrate a practical modern machine learning application of our Lie NAG-SC optimizer. The setting is a highly non-convex optimization problem with stochastic gradients, due to being a real deep learning task, but empirical success is still observed. More specifically, it was discovered [19] that adding artificial orthogonal constraints to attention layers in transformer models can improve their performances, because orthogonality disallows linearly dependent correlations between tokens, so that the learned attentions can be more efficient and robust. We will apply our optimizer to solve this constrained optimization problem.

The setup is the following (using the notation of [28]): consider a Scaled Dot-Product Multi-head Attention given by $\text{MultiHead}(Q, K, V) = \text{Concat}(\text{head}_1, \ldots, \text{head}_{n_{\text{head}}})W^O$, where $\text{head}_i = \text{Attention}(QW_i^Q, KW_i^K, VW_i^V)$, $\text{Attention}(\tilde{Q}, \tilde{K}, \tilde{V}) = \text{softmax}\left(\frac{\tilde{Q}\tilde{K}^\top}{\sqrt{d_k}}\right)\tilde{V}$. The trainable parameters are matrices $W_i^Q \in \mathbb{R}^{d_{\text{model}} \times d_k}$, $W_i^K \in \mathbb{R}^{d_{\text{model}} \times d_k}$, $W_i^V \in \mathbb{R}^{d_{\text{model}} \times d_v}$ and $W^O \in \mathbb{R}^{n_{\text{head}}d_v \times d_{\text{model}}}$. The

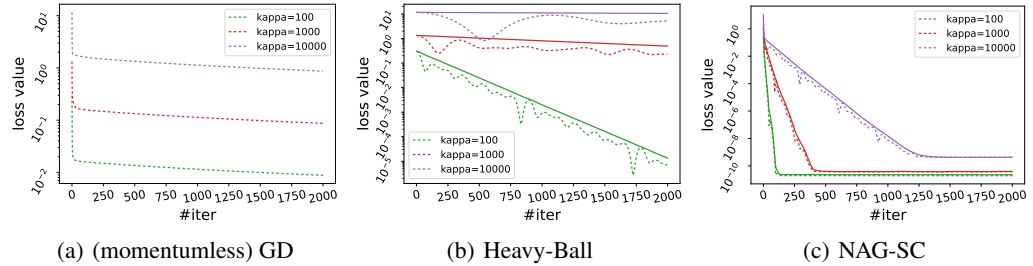

| (a) (momentumless) GD | (b) Heavy-Ball | (c) NAG-SC |

Figure 2: Local convergence of Lie Heavy-Ball and Lie NAG-SC on eigenvalue decomposition problem with different condition numbers. The initialization is close to the global minimum. The dashed curves are the value of potential function along the trajectory and the solid curves are the values of the corresponding Lyapunov functions. Lie GD (Eq. 1) has $h$ been chosen as $1/L$ [32, Thm. 15]. We observe: 1. Lie NAG-SC converges much faster than Lie Heavy-Ball, especially on ill-conditioned problems. 2. Although the potential function is not monotonely decreasing, the Lyapunov is.

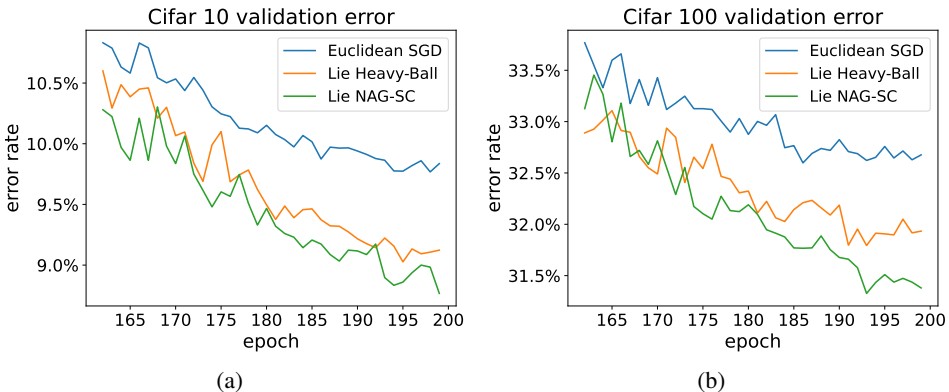

|   (a)   |   (b)   |

Figure 3: Training curve when applying our Lie HB and Lie NAG-SC to vision transformers. Forcing the query matrices and the key matrices on $\mathsf{SO}(n)$ [19, Orthogonality across heads] led to reduced training error. Moreover, validation error is also improved (Tab. 2).

three input matrices $Q$, $K$ and $V$ all have dimension $sequence\_length \times d_{\text{model}}$. $d_k$ and $d_v$ are usually smaller than $d_{\text{model}}$.

In the case $d_{\text{model}} = n_{\text{head}}d_k$, which is satisfied in many popular models, we apply the constraint 'orthogonality across heads' [19] and require $\text{Concat}(W_i^Q, i = 1..., n_{\text{head}})$ and $\text{Concat}(W_i^K, i = 1..., n_{\text{head}})$ to be in $\mathsf{SO}(d_{\text{model}})$. We compare the performance of our newly proposed optimizer with the existing ones (the optimizer in [19] is identical to Lie Heavy-Ball on $\mathsf{SO}(n)$). Fig. 3 and Tab. 2 are the validation error when we train a vision transformer [2] with 6.3M parameters from scratch on CIFAR, showing an improvement of Lie NAG-SC comparing the state-of-the-art algorithm Lie Heavy-Ball. The computations are done on a single Nvidia V100 GPU. The model structures and hyperparameters are identical as Sec. 3.2 in [19]. Each presented result is the average of 3 independent runs.

|  | Euclidean SGD | Lie HB | Lie NAG-SC |
|---|---|---|---|
| CIFAR 10 | 9.84% | 9.12% | 8.77% |
| CIFAR 100 | 32.68% | 31.93% | 31.38% |

Table 2: Validation error rate of vision transformer trained by different algorithms on CIFAR, showing the performance is ordered by Euclidean GD < Lie Heavy-Ball < Lie NAG-SC for both CIFAR 10 and 100. The blue font means the lowest error rate.

## Acknowledgments and Disclosure of Funding

The authors are grateful for the partially support by NSF DMS-1847802, Cullen-Peck Scholarship, and GT-Emory Humanity.AI Award. We thank the anonymous reviewers for their helpful comments.

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

# A  Properties of Lie groups and functions on Lie groups

## A.1  More details about compact Lie groups with left-invariant metric

Comparing with the Euclidean space, Lie groups lack of commutativity, i.e., for $g, \hat{g} \in \mathsf{G}$, $g\hat{g}$ and $\hat{g}g$ are not necessarily equal. This can also be characterized by the non-trivial Lie bracket $[\cdot, \cdot]$. This non-commutativity leads to the fact that $\exp(X)\exp(Y) \neq \exp(X + Y)$. An explicit expression for $\log(\exp(X)\exp(Y))$ is given by Dynkin's formula [13]. Utilizing Dynkin's formula, we quantify $\mathrm{d}\log$ in the following.

**Corollary 15** (Differential of logarithm)**.** *If $\mathrm{d}\log g$ is well defined, then the differential of logarithm on $\mathsf{G}$ is given by*

$$\mathrm{d}_\xi \log g := (\mathrm{d}\log)_g(T_e \mathsf{L}_g \xi) = T_e \mathsf{L}_g \left[ p(\mathrm{ad}_{\log g})\xi \right], \quad \forall \xi \in \mathfrak{g} \tag{17}$$

*where the power series $p$ is defined in Eq. (16)*

The vanishment of $\mathrm{ad}_\xi^* \xi$ can also be understood as the group structure and the Riemannian structure are compatible. See [18] for more discussion. Under such assumption, we have the following properties:

**Corollary 16.** *Suppose we have $\mathrm{ad}_X$ is skew-adjoint $\forall X \in \mathfrak{g}$. Then for any $g \in \mathsf{G}$ and any $\xi \in \mathfrak{g}$ such that $\mathrm{d}_\xi \log g$ is well-defined, we have*

$$\langle \mathrm{d}_\xi \log g, \log g \rangle = \langle \log g, \xi \rangle$$

**Corollary 17.** *When $\mathrm{d}_\xi \log g$ is well-defined and $\mathrm{ad}_X$ is skew-adjoint $\forall X \in \mathfrak{g}$, we have*

$$\langle \mathrm{d}_\xi \log g, \xi \rangle \leq \|\xi\|^2$$

**Corollary 18.** *Define*

$$A := \max_{\|X\|=1} \|\mathrm{ad}_X\|_{op} \tag{18}$$

*When $d(g, e) \leq \frac{a}{A}$ for some $a \in (0, 2\pi)$, we have*

$$\|\mathrm{d}\log g - Id\| \leq q(a)$$

*where $q$ is defined by*

$$
\begin{aligned}
q(x) &:= \|p(xi) - 1\| \\
&= \left\| \frac{xi}{1 - \cos x + i\sin x} - 1 \right\| \\
&= \sqrt{\frac{1 - \frac{x^2}{2} - \cos x - x\sin x}{1 - \cos x}}
\end{aligned}
\tag{19}
$$

*with $p$ defined in Eq. (16).*

**Remark 19** (About existence and uniqueness of $\log$)**.** *As the inverse of $\exp$, the operator $\log$ may not be uniquely defined globally. However, we are always considering in a unique geodesic subset of the Lie group, where $\log$ is defined uniquely in such a subset of Lie group. Similarly, even if we do not have globally geodesically strongly convex functions, we only require locally strong convexity.*

## A.2  More details about functions on Lie groups

The commonly used geodesic $L$-smooth on a manifold $M$ is given by the following definition [e.g., 32, Def. 5]:

**Definition 20** (Geodesically $L$-smooth)**.** *$U : \mathsf{G} \to \mathbb{R}$ is geodesically $L$-smooth if for any $g, \hat{\in} M$,*

$$\left\| \nabla U(g) - \Gamma_{\hat{g}}^g \nabla U(\hat{g}) \right\| \leq L d(g, \hat{g})$$

*where $\Gamma_{\hat{g}}^g$ is the parallel transport from $\hat{g}$ to $g$.*

**Lemma 21.** *Under the assumption of $\mathrm{ad}_X^*$ is skew-adjoint $\forall X \in \mathfrak{g}$, Def. 4 is identical to Def. 20.*

*Proof of Lemma 21.* For any $g, \hat{g} \in \mathsf{G}$, consider the shortest geodesic $\phi : [0,1] \to \mathsf{G}$ connecting $g$ and $\hat{g}$ and denote $\xi = T_g \mathsf{L}_{g^{-1}} \nabla U(g)$. Using the condition $\mathrm{ad}$ is skew-adjoint, we have $\dot{\phi}(t) = 0$ and $T_e \mathsf{L}_{\phi(t)} \xi$ is parallel along $\phi$ by checking the condition for parallel transport [14, Thm. 1]:

$$\frac{d}{dt}\xi = 0 = -\frac{1}{2}\left[T_{\phi(t)} \mathsf{L}_{\phi(t)^{-1}} \dot{\phi}(t), \xi\right]$$

This tells that

$$T_g \mathsf{L}_{g^{-1}} \Gamma_g^{\hat{g}} \nabla f(g) = T_{\hat{g}} \mathsf{L}_{\hat{g}^{-1}} \nabla f(\hat{g})$$

Together with the metric is left-invariant, we have

$$\left\| T_g \mathsf{L}_{g^{-1}} \nabla U(g) - T_{\hat{g}} \mathsf{L}_{\hat{g}^{-1}} \nabla U(\hat{g}) \right\| \le L d(g, \hat{g})$$

which is identical to Def. 4. $\qquad\square$

**Corollary 22** (Properties of $L$-smooth functions). *If $U : \mathsf{G} \to \mathbb{R}$ is $L$-smooth, then for any $g, \hat{g} \in \mathsf{G}$, we have*

$$U(\hat{g}) \le U(g) + \left\langle T_g \mathsf{L}_{g^{-1}} \nabla U(g), \log g^{-1}\hat{g} \right\rangle + \frac{L}{2} d^2(g, \hat{g}) \tag{20}$$

*Proof of Cor. 22.* We denote the one of the shortest geodesic connecting $g$ and $\hat{g}$ as $g(t)$, i.e., $\pi : [0,1] \to \mathsf{G}$ with $g(0) = g$ and $g(1) = \hat{g}$, $g(t) = g \exp(t\xi)$ for some $\xi \in \mathfrak{g}$ with $\|\xi\| = d(g, \hat{g})$. Then

$$U(\hat{g}) - U(g)$$

$$= \left\langle T_e \mathsf{L}_g \xi, \nabla U(g) \right\rangle + \int_0^1 \left\langle T_e \mathsf{L}_{g(t)} \xi, \nabla U(g(t)) \right\rangle \mathrm{d}t$$

$$= \left\langle \xi, T_g \mathsf{L}_{g^{-1}} \nabla U(g) \right\rangle + \int_0^1 \left\langle \xi, T_{g(t)} \mathsf{L}_{g(t)^{-1}} \nabla U(g(t)) - T_g \mathsf{L}_{g^{-1}} \nabla U(g) \right\rangle \mathrm{d}t$$

$$\le \left\langle \xi, T_g \mathsf{L}_{g^{-1}} \nabla U(g) \right\rangle + \int_0^1 \|\xi\| \left\| T_{g(t)} \mathsf{L}_{g(t)^{-1}} \nabla U(g(t)) - T_g \mathsf{L}_{g^{-1}} \nabla U(g) \right\| \mathrm{d}t$$

$$\le \left\langle \xi, T_g \mathsf{L}_{g^{-1}} \nabla U(g) \right\rangle + \int_0^1 tL\|\xi\|^2 \, \mathrm{d}t$$

$$= \left\langle \xi, T_g \mathsf{L}_{g^{-1}} \nabla U(g) \right\rangle + \frac{L}{2} d^2(g, \hat{g})$$

$\qquad\square$

**Lemma 23** (Co-coercivity). *If the function $U : \mathsf{G} \to \mathbb{R}$ is both $L$-smooth and convex on a geodesically convex set $S \subset \mathsf{G}$, then we have for any $g, \hat{g} \in S$,*

$$U(\hat{g}) \ge U(g) + \left\langle T_g \mathsf{L}_{g^{-1}} \nabla U(g), \log g^{-1}\hat{g} \right\rangle + \frac{1}{2L} \left\| T_g \mathsf{L}_{g^{-1}} \nabla U(g) - T_{\hat{g}} \mathsf{L}_{\hat{g}^{-1}} \nabla U(\hat{g}) \right\|^2 \tag{21}$$

*Proof of Lemma 23.* By convexity, we have for any $g \in \mathsf{G}$, $\xi \in \mathfrak{g}$ and $t \in [0,1]$,

$$U(g\exp(t\xi)) - U(g) \ge t\left\langle T_g \mathsf{L}_{g^{-1}} \nabla U(g), \xi \right\rangle$$

$$U(g) - U(g\exp(t\xi)) \ge -t\left\langle T_{g\exp(t\xi)} \mathsf{L}_{g\exp(t\xi)^{-1}} \nabla U(g\exp(t\xi)), \xi \right\rangle$$

We sum these two inequalities and have

$$\left\langle T_{g\exp(t\xi)} \mathsf{L}_{g\exp(t\xi)^{-1}} \nabla U(g\exp(t\xi)) - T_g \mathsf{L}_{g^{-1}} \nabla U(g), \xi \right\rangle \ge 0$$

which tells that

$$\frac{\partial}{\partial t}\left[ T_{g\exp(t\xi)} \mathsf{L}_{g\exp(t\xi)^{-1}} \nabla U(g\exp(t\xi)) \right] = H_t \xi$$

for some linear map $H : \mathfrak{g} \to \mathfrak{g}$ with all eigenvalues between 0 and $L$, i.e., $0 \le H_t \le L$ for any $t \in [0,1]$.

Now, we select the shortest geodesic connection $g$ and $\hat{g}$, defined by $g(t) := g\exp(t\xi)$, with $\hat{g} = g\exp(\xi)$ and $\|\xi\| = d(g, \hat{g})$. By

$$T_{g(t)}\mathsf{L}_{g(t)^{-1}}\nabla U(g(t)) - T_g\mathsf{L}_{g^{-1}}\nabla U(g) = \int_0^t \frac{\partial}{\partial s}T_{g\exp(s\xi)}\mathsf{L}_{g\exp(s\xi)^{-1}}\nabla U(g\exp(s\xi))ds$$

$$= \int_0^t H_s\xi ds$$

we have

$$U(\hat{g}) - U(g)$$

$$= \int_0^1 \langle T_{g(t)}\nabla U(g(t)), \xi\rangle$$

$$= \int_0^1 \Big\langle T_g\mathsf{L}_{g^{-1}}\nabla U(g) + \int_0^t \frac{\partial}{\partial s}T_{g(s)}\mathsf{L}_{g(s)^{-1}}\nabla U(g(s))ds, \xi\Big\rangle dt$$

$$= \langle T_g\mathsf{L}_{g^{-1}}\nabla U(g), \xi\rangle + \int_0^1 \Big\langle \int_0^t H_s\xi ds, \xi\Big\rangle dt$$

$$= \langle T_g\mathsf{L}_{g^{-1}}\nabla U(g), \xi\rangle + \int_0^1 \int_0^t \langle H_s\xi, \xi\rangle ds dt$$

$$= \langle T_g\mathsf{L}_{g^{-1}}\nabla U(g), \xi\rangle + \frac{1}{2}\int_0^1 \int_0^1 \langle H_s\xi, \xi\rangle ds dt$$

$$\geq \langle T_g\mathsf{L}_{g^{-1}}\nabla U(g), \xi\rangle + \frac{1}{2L}\int_0^1 \int_0^1 \langle H_s\xi, H_s\xi\rangle ds dt$$

$$\geq \langle T_g\mathsf{L}_{g^{-1}}\nabla U(g), \xi\rangle + \frac{1}{2L}\int_0^1 \Big(\int_0^1 H_s\xi ds\Big)^2 dt$$

$$= \langle T_g\mathsf{L}_{g^{-1}}\nabla U(g), \xi\rangle + \frac{1}{2L}\int_0^1 \Big\|T_{\hat{g}}\mathsf{L}_{\hat{g}^{-1}}\nabla U(\hat{g}) - T_g\mathsf{L}_{g^{-1}}\nabla U(g)\Big\|^2 dt$$

$$= \langle T_g\mathsf{L}_{g^{-1}}\nabla U(g), \xi\rangle + \frac{1}{2L}\Big\|T_{\hat{g}}\mathsf{L}_{\hat{g}^{-1}}\nabla U(\hat{g}) - T_g\mathsf{L}_{g^{-1}}\nabla U(g)\Big\|^2$$

$\square$

**Corollary 24.** *If the function $U : \mathsf{G} \to \mathbb{R}$ is both $L$-smooth and convex on a geodesically convex set $S \subset \mathsf{G}$, then we have for any $g, \hat{g} \in S$,*

$$\langle T_g\mathsf{L}_{g^{-1}}\nabla U(g) - T_{\hat{g}}\mathsf{L}_{\hat{g}^{-1}}\nabla U(\hat{g}), \log \hat{g}^{-1}g\rangle \geq \frac{1}{L}\Big\|T_g\mathsf{L}_{g^{-1}}\nabla U(g) - T_{\hat{g}}\mathsf{L}_{\hat{g}^{-1}}\nabla U(\hat{g})\Big\|^2 \quad (22)$$

*Proof.* By exchanging $g$ and $\hat{g}$ in Eq. (21), we have

$$U(g) \geq U(\hat{g}) + \langle T_{\hat{g}}\mathsf{L}_{\hat{g}^{-1}}\nabla U(\hat{g}), \log \hat{g}^{-1}g\rangle + \frac{1}{2L}\Big\|T_g\mathsf{L}_{g^{-1}}\nabla U(g) - T_{\hat{g}}\mathsf{L}_{\hat{g}^{-1}}\nabla U(\hat{g})\Big\|^2 \quad (23)$$

Summing up 21 and 23 gives us the desired result. $\square$

**Corollary 25** (Properties of $\mu$-strongly convex functions)**.** *Suppose $U : \mathsf{G} \to \mathbb{R}$ is geodesic-$\mu$-strongly convex on a geodesically convex set $S \subset \mathsf{G}$, then for any $g, \hat{g} \in S$,*

$$\langle T_g\mathsf{L}_{g^{-1}}\nabla U(g), \log \hat{g}^{-1}g\rangle \geq \mu d^2(g, \hat{g}) \quad (24)$$

*Proof of Cor. 25.* By the definition of geodesic-$\mu$-strongly convex functions in Eq. (4), we have

$$U(g) - U(\hat{g}) \geq \langle T_{\hat{g}}\mathsf{L}_{\hat{g}^{-1}}\nabla U(\hat{g}), \log \hat{g}^{-1}g\rangle + \frac{\mu}{2}\Big\|\log \hat{g}^{-1}g\Big\|^2$$

$$U(\hat{g}) - U(g) \geq \langle T_g\mathsf{L}_{g^{-1}}\nabla U(g), \log g^{-1}\hat{g}\rangle + \frac{\mu}{2}\Big\|\log \hat{g}^{-1}g\Big\|^2$$

Summing them up and using $\log g^{-1}\hat{g} = -\log \hat{g}^{-1}g$ gives us the conclusion. $\square$

# B  More details about optimization ODE

*Proof of Thm. 6.* By direct calculation,

$$\frac{d}{dt}E(g(t),\xi(t)) = \langle T_e \mathsf{L}_g \xi, \nabla U(g) \rangle + \langle \xi, \dot{\xi} \rangle$$
$$= -\gamma \|\xi\|^2$$

$\square$

**Lemma 26** (Monotonicity of the Lyapunov function). *Assume* $\mathrm{ad}_X$ *is skew-adjoint for any* $X \in \mathfrak{g}$. *Suppose there is a geodesically convex set* $S \subset \mathsf{G}$ *satisfying:*

- *$U$ is geodesic-$\mu$-strongly convex on a geodesically convex set $S \subset \mathsf{G}$.*

- *$g_*$ is the minimum of $U$ on $S$.*

- *$g(t) \in S$ for all $t \geq 0$.*

- *$\log g_*^{-1} g$ and its differential is well-defined for all $g \in S$.*

*Then the solution of ODE* (2) *satisfies*

$$\frac{d}{dt}\mathcal{L}^{ODE}(g(t),\xi(t)) \leq -c_{ODE}\mathcal{L}^{ODE}(g(t),\xi(t)) \tag{25}$$

*with the convergence rate given by*

$$c_{ODE} := \gamma \min\left\{\frac{\mu}{\mu + \gamma^2}, \frac{2}{3}\right\}$$

*Proof of Lemma 26.* The time derivative of $L$ is

$$\frac{d}{dt}\mathcal{L}^{\mathrm{ODE}} = \langle \xi, T_g \mathsf{L}_{g^{-1}} \nabla U(g) \rangle + \frac{1}{2}\langle \xi, -\gamma\xi - T_g \mathsf{L}_{g^{-1}} \nabla U(g) \rangle$$
$$+ \frac{1}{2}\langle \gamma \log g_*^{-1} g + \xi, \gamma \, \mathrm{d}_\xi \log g_*^{-1} g - \gamma\xi - T_g \mathsf{L}_{g^{-1}} \nabla U(g) \rangle$$
$$= \langle \xi, T_g \mathsf{L}_{g^{-1}} \nabla U(g) \rangle - \frac{\gamma}{2}\langle \xi, \xi \rangle - \frac{1}{2}\langle \xi, T_g \mathsf{L}_{g^{-1}} \nabla U(g) \rangle$$
$$+ \frac{\gamma^2}{2}\langle \log g_*^{-1} g, \mathrm{d}_\xi \log g_*^{-1} g \rangle - \frac{\gamma^2}{2}\langle \log g_*^{-1} g, \xi \rangle - \frac{\gamma}{2}\langle \log g_*^{-1} g, T_g \mathsf{L}_{g^{-1}} \nabla U(g) \rangle$$
$$+ \frac{\gamma}{2}\langle \xi, \mathrm{d}_\xi \log g_*^{-1} g \rangle - \frac{\gamma}{2}\langle \xi, \xi \rangle - \frac{1}{2}\langle \xi, \nabla U(g) \rangle$$
$$= -\gamma\langle \xi, \xi \rangle - \frac{\gamma}{2}\langle \log g_*^{-1} g, T_g \mathsf{L}_{g^{-1}} \nabla U(g) \rangle + \frac{\gamma}{2}\langle \xi, \mathrm{d}_\xi \log g_*^{-1} g \rangle$$
$$\leq -\frac{\gamma}{2}\langle \log g_*^{-1} g, T_g \mathsf{L}_{g^{-1}} \nabla U(g) \rangle - \frac{\gamma}{2}\langle \xi, \xi \rangle$$

where the second last equation is by Cor. 16 and the last inequality is by Cor. 17. By the property of strong convexity in Eq. (4) and Cor. 25, for any $\lambda \in [0, 1]$,

$$\frac{d}{dt}\mathcal{L}^{\mathrm{ODE}} \leq -\frac{\gamma}{2}\left(\lambda(U(g) - U(g_*)) + \left(\frac{\lambda}{2} + (1-\lambda)\right)\mu d^2(g_*, g) + \|\xi\|^2\right) \tag{26}$$

where $\lambda$ is a constant to be determined later.

Next, we try to give $\mathcal{L}^{\mathrm{ODE}}$ an upper bound. By Cauchy-Schwarz inequality,

$$\left\|\gamma \log g_*^{-1} g + \xi\right\|^2 \leq 2\left(\gamma^2 d^2(g_*, g) + \|\xi\|^2\right)$$

and further

$$\mathcal{L}^{\mathrm{ODE}} \leq U(g) - U(g_*) + \frac{3}{4}\|\xi\|^2 + \frac{\gamma^2}{2}d^2(g, g_*) \tag{27}$$

| | Splitting scheme | Heavy ball |
|---|---|---|
| velocity | $\xi$ | $\sqrt{\frac{h\gamma}{1-e^{-\gamma h}}}\xi$ |
| friction parameter | $\gamma$ | $\sqrt{\frac{\gamma(1-e^{-\gamma h})}{h}}$ |
| step size | $h$ | $\sqrt{\frac{1-e^{-\gamma h}}{h\gamma}}h$ |

Table 3: Change of variable between Heavy-ball and splitting scheme

Compare the coefficients in Eq. (26) and (27), we have

$$\frac{d}{dt}\mathcal{L}^{\text{ODE}} \leq -c_{\text{ODE}}\mathcal{L}^{\text{ODE}}$$

where the convergence rate $c_{\text{ODE}}$ is given by

$$c_{\text{ODE}} := \frac{\gamma}{2}\min\left\{\lambda, \frac{4}{3}, \frac{2}{\gamma^2}\left(\frac{\lambda}{2} + (1-\lambda)\right)\mu\right\}$$

$$= \frac{\gamma}{2}\min\left\{\lambda, \frac{4}{3}, \frac{\mu}{\gamma^2}(2-\lambda)\right\}$$

By selecting $\lambda = \frac{2\mu}{\mu+\gamma^2}$ to make $\lambda = \frac{\mu}{\gamma^2}(2-\lambda)$ satisfied, we have the desired result. $\quad\square$

*Proof of Thm. 9.* This is the direct corollary from Cor. 8 and Lemma 26. $\quad\square$

## C   More details about Heavy-Ball discretization

**Remark 27** (Polyak's Heavy ball [23]). *Heavy-ball scheme in the Euclidean space is*

$$x_{k+1} = x_k + \alpha(x_k - x_{k-1}) - \beta\nabla U(x_k)$$

*where $\alpha$ and $\beta$ are positive parameters. We now write it into a position-velocity form. By setting $v_{k+1} = \frac{x_{k+1}-x_k}{\sqrt{\beta}}$, we have*

$$\begin{cases} x_{k+1} = x_k + \sqrt{\beta}v_{k+1} \\ v_{k+1} = \alpha v_k - \sqrt{\beta}\nabla U(x_k) \end{cases}$$

*We perform the change of variables given by $\sqrt{\beta} \to h$, $\alpha \to 1 - \gamma h$ gives*

$$\begin{cases} x_{k+1} = x_k + hv_{k+1} \\ v_{k+1} = (1-\gamma h)v_k - h\nabla U(x_k) \end{cases}$$

*which is the Euclidean version corresponding to Eq. (9).*

**Remark 28** (Splitting discretization). *The two systems of ODEs in Eq. (8) are both linear and has exact solutions. We will refer to the numerical scheme evolving their exact solution alternatively by splitting discretization. More precisely, this gives us the following numerical scheme:*

$$\begin{cases} g_{k+1} = g_k\exp(h\xi_{k+1}) \\ \xi_{k+1} = e^{-\gamma h}\xi_k - \frac{1-e^{-\gamma h}}{\gamma}T_{g_k}\mathsf{L}_{g_k^{-1}}\nabla U(g_k) \end{cases} \tag{28}$$

*After change of variable in the Table 3, it becomes Eq. (9).*

*Eq. (28) is similar to the 'NAG-SC' in [26]. The authors provide a second-order approximation to the optimization ODE Eq. (2) by evolving the two ODEs in Eq. (8) in the following way in each step: 1) evolve $\xi$-ODE for $h/2$ time; 2) evolve $g$-ODE for $h/2$ time; 3) evolve $\xi$-ODE for $h/2$ time again. Although this zig-zag scheme is higher-order of approximation of the optimization ODE comparing to the the splitting approximation mentioned in Rmk. 28, it still has the same condition number dependence. The reason is, we can take out the first evolution of time $h/2$ for the $\xi$-system and it becomes identical to the splitting scheme (with a different initial condition.)*

Before we start the theoretical calculation, we mention that update for $\xi$ in Heavy-Ball scheme Eq. (9) can also be written as

$$\xi_k = \frac{1}{1-\gamma h}\xi_{k+1} + \frac{h}{1-\gamma h}T_{g_k}\mathsf{L}_{g_k^{-1}}\nabla U(g_k) \tag{29}$$

which will be helpful later.

*Proof of Thm. 11.* Using Eq. (29), we have the following calculation of $E^{\text{HB}}(g_k,\xi_k) - E^{\text{HB}}(g_{k-1},\xi_{k-1})$:

$$E^{\text{HB}}(g_k,\xi_k) - E^{\text{HB}}(g_{k-1},\xi_{k-1})$$

$$= U(g_k) - U(g_{k-1}) + \frac{(1-\gamma h)^2}{2}\left(\|\xi_k\|^2 - \|\xi_{k-1}\|^2\right)$$

$$= U(g_{k-1}\exp(h\xi_k)) - U(g_{k-1}) + \frac{(1-\gamma h)^2}{2}\left(\|\xi_k\|^2 - \left\|\frac{1}{1-\gamma h}\xi_k + \frac{h}{1-\gamma h}T_{g_{k-1}}\mathsf{L}_{g_{k-1}^{-1}}\nabla U(g_{k-1})\right\|^2\right)$$

$$= U(g_{k-1}\exp(h\xi_k)) - U(g_{k-1}) - \left(\gamma h - \frac{\gamma^2 h^2}{2}\right)\|\xi_k\|^2 - h\langle\xi_k, T_{g_{k-1}}\mathsf{L}_{g_{k-1}^{-1}}\nabla U(g_{k-1})\rangle - \frac{h^2}{2}\|\nabla U(g_{k-1})\|^2$$

$$\leq h\langle\xi_k, T_{g_{k-1}}\mathsf{L}_{g_{k-1}^{-1}}\nabla U(g_{k-1})\rangle + \frac{Lh^2}{2}\|\xi_k\|^2 - \left(\gamma h - \frac{\gamma^2 h^2}{2}\right)\|\xi_k\|^2 - h\langle\xi_k, T_{g_{k-1}}\mathsf{L}_{g_{k-1}^{-1}}\nabla U(g_{k-1})\rangle - \frac{h^2}{2}\|\nabla U(g_{k-1})\|^2$$

$$\leq \frac{1}{2}(h^2 L - 2\gamma h + \gamma^2 h^2)\|\xi_k\|^2 - \frac{h^2}{2}\|\nabla U(g_{k-1})\|^2$$

where the second last inequality is the property of $L$-smooth functions given in Eq. 20.

When $h \leq \frac{\gamma}{\gamma^2 + L}$, we have $h^2 L - 2\gamma h + \gamma^2 h^2 \leq -\gamma h$, and $E^{\text{HB}}(g_k,\xi_k) - E^{\text{HB}}(g_{k-1},\xi_{k-1}) \leq -\gamma h\|\xi_k\|^2$. □

**Remark 29.** *The design of modified energy for Heavy-Ball in Eq. (10) and Thm. 11 is new, and is different from modified potential function in existing works (e.g., [15]). Our modified energy is not designed to let the Hamiltonian system to have higher order of preserving the total energy, but is defined to ensure monotonicity of the modified energy to ensure global convergence of the numerical scheme.*

**Remark 30.** *We specially choose our update of numerical scheme in Eq. (9) (and also later Eq. (13) for NAG-SC) to ensure such natural form of the modified energy. If we choose to update g first and then $\xi$ (e.g., [25]), the modified energy will have to be evaluated at different time step, $E^{HB}(g_{k+1},\xi_k)$, for example.*

*Proof of Cor. 12.* We prove this by induction. Suppose we have $\mathfrak{g}_k \in S_0$. By the dissipation of the modified energy Thm. 11, $E^{\text{HB}}(g_k,\xi_k) \leq E^{\text{HB}}(g_0,\xi_0)$. As a result,

$$\|\xi_{k+1}\| \leq (1-\gamma h)\|\xi_k\| + h\|\nabla U(g_k)\|$$

$$\leq (1-\gamma h)\sqrt{\frac{2}{(1-\gamma h)^2}E^{\text{HB}}(g_k,\xi_k)} + h\|\nabla U(g_k)\|$$

$$\leq \sqrt{2E^{\text{HB}}(g_0,\xi_0)} + h\max_{S_0}\|\nabla U\|$$

Since we have $g_{k+1} = g_k\exp(h\xi_{k+1})$, we have $d(g_{k+1},g_k) \leq h\|\xi_{k+1}\|$. Together with the condition that $U(g_{k+1}) \leq E^{\text{HB}}(g_{k+1},\xi_{k+1}) \leq u$, we have $g_{k+1} \in S_0$. Mathematical induction gives the desired result. □

**Lemma 31.** *Assume* $\text{ad}_X$ *is skew-adjoint for any* $X \in \mathfrak{g}$. *Suppose there is a geodesically convex set* $S \subset \mathsf{G}$ *satisfying:*

- *U is geodesically $\mu$-strongly convex on a convex set $S \subset \mathsf{G}$.*

- *$g_*$ is the minima of $U$ on $S$.*

- $g_k \in S$ for all $k \in \mathbb{N}$.

- $\log g_*^{-1} g$ and its differential is well-defined for all $g \in S$.

*Then we have*

$$\mathcal{L}_{k+1}^{HB} - \mathcal{L}_k^{HB} \leq -b_{HB}\mathcal{L}_{k+1}^{HB} - \frac{h}{2(1-\gamma h)}\left(\frac{3\gamma}{4L} - \frac{h}{1-\gamma h}\right)\|\nabla U(g_{k+1})\|^2$$

*where $b_{HB}$ is given by*

$$b_{HB} := \min\left\{\frac{\gamma h}{8}, \frac{\mu h(1-\gamma h)}{2\gamma}, \frac{2\gamma h}{3(1-\gamma h)}\right\}$$

*Proof of Lemma 31.* For $(g_k, \xi_k)$ following the Heavy-Ball scheme Eq. (9), we use the shorthand notation of $\mathcal{L}_k^{HB} := \mathcal{L}^{HB}(g_k, \xi_k)$, which gives

$$\mathcal{L}_k^{HB} = \frac{1}{1-\gamma h}\left(U(g_{k-1}) - U(g_*)\right) + \frac{1}{4}\|\xi_k\|^2 + \frac{1}{4}\left\|\frac{\gamma}{1-\gamma h}\log g_*^{-1}g_k + \xi_k\right\|^2 \qquad (30)$$

Evaluate of the three terms in $\mathcal{L}_{k+1}^{HB} - \mathcal{L}_k^{HB}$ separately :

•The first term: By co-coercivity in Lemma 23, we have

$$\frac{1}{1-\gamma h}\left(U(g_k) - U(g_{k-1})\right)$$

$$\leq \frac{h}{1-\gamma h}\left\langle T_{g_k}\mathsf{L}_{g_k^{-1}}\nabla U(g_k), \xi_k\right\rangle \qquad\qquad I_1$$

$$- \frac{1}{2L}\frac{1}{1-\gamma h}\left\|T_{g_k}\mathsf{L}_{g_k^{-1}}\nabla U(g_k) - T_{g_{k-1}}\mathsf{L}_{g_{k-1}^{-1}}\nabla U(g_{k-1})\right\|^2 \qquad\qquad I_2$$

•The second term: Using Eq. (29), we have

$$\frac{1}{4}\left(\|\xi_{k+1}\|^2 - \|\xi_k\|^2\right)$$

$$= \frac{1}{2}\langle\xi_{k+1} - \xi_k, \xi_{k+1}\rangle - \frac{1}{4}\|\xi_{k+1} - \xi_k\|^2$$

$$= -\frac{\gamma h}{2(1-\gamma h)}\|\xi_{k+1}\|^2 \qquad\qquad II_1$$

$$- \frac{h}{2(1-\gamma h)}\left\langle\xi_{k+1}, T_{g_k}\mathsf{L}_{g_k^{-1}}L_\nabla U(g_k)\right\rangle \qquad\qquad II_2$$

$$- \frac{1}{4}\|\xi_{k+1} - \xi_k\|^2 \qquad\qquad II_3$$

•The third term: Define a parametric curve $[0,h] \to \mathsf{G}$ as $g_t := g_k\exp(t\xi_{k+1})$

$$\frac{1}{4}\left(\left\|\frac{\gamma}{1-\gamma h}\log g_*^{-1}g_{k+1} + \xi_{k+1}\right\|^2 - \left\|\frac{\gamma}{1-\gamma h}\log g_*^{-1}g_k + \xi_k\right\|^2\right)$$

$$= \frac{1}{4}\left(\left\|\frac{\gamma}{1-\gamma h}\log g_*^{-1}g_{k+1} + \xi_{k+1}\right\|^2 - \left\|\frac{\gamma}{1-\gamma h}\log g_*^{-1}g_k + \xi_{k+1}\right\|^2\right)$$

$$+ \frac{1}{4}\left(\left\|\frac{\gamma}{1-\gamma h}\log g_*^{-1}g_k + \xi_{k+1}\right\|^2 - \left\|\frac{\gamma}{1-\gamma h}\log g_*^{-1}g_k + \xi_k\right\|^2\right)$$

$$= \frac{1}{2}\int_0^h\left\langle\frac{d}{dt}\left(\frac{\gamma}{1-\gamma h}\log g_*^{-1}g_t + \xi_{k+1}\right), \frac{\gamma}{1-\gamma h}\log g_*^{-1}g_t + \xi_{k+1}\right\rangle dt$$

$$+ \frac{1}{4}\left(\left\|\frac{\gamma}{1-\gamma h}\log g_*^{-1}g_k + \xi_{k+1}\right\|^2 - \left\|\frac{\gamma}{1-\gamma h}\log g_*^{-1}g_k + \xi_k\right\|^2\right)$$

We evaluate the two terms separately.

$$\int_0^h \left\langle \frac{d}{dt}\left( \frac{\gamma}{1-\gamma h}\log g_*^{-1}g_t + \xi_{k+1} \right), \frac{\gamma}{1-\gamma h}\log g_*^{-1}g_t + \xi_{k+1} \right\rangle dt$$

$$= \int_0^h \left\langle \frac{\gamma}{1-\gamma h}\,d_{\xi_{k+1}}\log g_*^{-1}g_t, \frac{\gamma}{1-\gamma h}\log g_*^{-1}g_t + \xi_{k+1} \right\rangle dt$$

$$= \int_0^h \left\langle \frac{\gamma}{1-\gamma h}\,d_{\xi_{k+1}}\log g_*^{-1}g_t, \frac{\gamma}{1-\gamma h}\log g_*^{-1}g_t \right\rangle dt$$

$$+ \int_0^h \left\langle \frac{\gamma}{1-\gamma h}\,d_{\xi_{k+1}}\log g_*^{-1}g_t, \xi_{k+1} \right\rangle dt$$

$$\leq \frac{\gamma^2}{(1-\gamma h)^2}\int_0^h \left\langle \xi_{k+1}, \log g_*^{-1}g_t \right\rangle dt + \frac{\gamma h}{1-\gamma h}\|\xi_{k+1}\|^2$$

$$= \frac{\gamma^2}{(1-\gamma h)^2}\int_0^h \left\langle \xi_{k+1}, \log g_*^{-1}g_t - \log g_*^{-1}g_k \right\rangle dt + \frac{\gamma^2 h}{(1-\gamma h)^2}\left\langle \xi_{k+1}, \log g_*^{-1}g_k \right\rangle + \frac{\gamma h}{1-\gamma h}\|\xi_{k+1}\|^2$$

$$= \frac{\gamma^2}{(1-\gamma h)^2}\int_0^h \left\langle \xi_{k+1}, \int_0^t d_{\xi_{k+1}}\log g_*^{-1}g_s\,ds \right\rangle dt + \frac{\gamma^2 h}{(1-\gamma h)^2}\left\langle \xi_{k+1}, \log g_*^{-1}g_k \right\rangle + \frac{\gamma h}{1-\gamma h}\|\xi_{k+1}\|^2$$

$$= \frac{\gamma^2}{(1-\gamma h)^2}\int_0^h \int_0^t \left\langle \xi_{k+1}, d_{\xi_{k+1}}\log g_*^{-1}g_s \right\rangle ds\,dt + \frac{\gamma^2 h}{(1-\gamma h)^2}\left\langle \xi_{k+1}, \log g_*^{-1}g_k \right\rangle + \frac{\gamma h}{1-\gamma h}\|\xi_{k+1}\|^2$$

$$\leq \frac{\gamma^2}{(1-\gamma h)^2}\int_0^h \int_0^t \|\xi_{k+1}\|^2 ds\,dt + \frac{\gamma^2 h}{(1-\gamma h)^2}\left\langle \xi_{k+1}, \log g_*^{-1}g_k \right\rangle + \frac{\gamma h}{1-\gamma h}\|\xi_{k+1}\|^2$$

$$= \frac{\gamma^2 h^2}{2(1-\gamma h)^2}\|\xi_{k+1}\|^2 + \frac{\gamma^2 h}{(1-\gamma h)^2}\left\langle \xi_{k+1}, \log g_*^{-1}g_k \right\rangle + \frac{\gamma h}{1-\gamma h}\|\xi_{k+1}\|^2$$

Using the $\xi$ update in Eq. (29), we have

$$\left\| \frac{\gamma}{1-\gamma h}\log g_*^{-1}g_{k+1} + \xi_{k+1} \right\|^2 - \left\| \frac{\gamma}{1-\gamma h}\log g_*^{-1}g_k + \xi_k \right\|^2$$

$$= \left\langle \xi_{k+1} - \xi_k, \frac{2\gamma}{1-\gamma h}\log g_*^{-1}g_k + \xi_{k+1} + \xi_k \right\rangle$$

$$= -\left\langle \frac{\gamma h}{1-\gamma h}\xi_{k+1} + \frac{h}{1-\gamma h}T_{g_k}\mathsf{L}_{g_k^{-1}}\nabla U(g_k), \frac{2\gamma}{1-\gamma h}\log g_*^{-1}g_k + \frac{2-\gamma h}{1-\gamma h}\xi_{k+1} + \frac{h}{1-\gamma h}T_{g_k}\mathsf{L}_{g_k^{-1}}\nabla U(g_k) \right\rangle$$

$$= -\frac{2\gamma^2 h}{(1-\gamma h)^2}\left\langle \xi_{k+1}, \log g_*^{-1}g_k \right\rangle - \frac{\gamma h(2-\gamma h)}{(1-\gamma h)^2}\|\xi_{k+1}\|^2 - \frac{2\gamma h}{(1-\gamma h)^2}\left\langle T_{g_k}\mathsf{L}_{g_k^{-1}}\nabla U(g_k), \log g_*^{-1}g_k \right\rangle$$

$$- \frac{2h}{(1-\gamma h)^2}\left\langle T_{g_k}\mathsf{L}_{g_k^{-1}}\nabla U(g_k), \xi_{k+1} \right\rangle - \frac{h^2}{(1-\gamma h)^2}\|\nabla U(g_k)\|^2$$

Sum them up, we have

$$\frac{1}{4}\left( \left\| -\frac{\gamma}{1-\gamma h}\log g_{k+1}^{-1}g_* + \xi_{k+1} \right\|^2 - \left\| -\frac{\gamma}{1-\gamma h}\log g_k^{-1}g_* + \xi_k \right\|^2 \right)$$

$$\leq -\frac{h}{2(1-\gamma h)^2}\left\langle T_{g_k}\mathsf{L}_{g_k^{-1}}\nabla U(g_k), \xi_{k+1} \right\rangle \qquad\qquad III_2$$

$$- \frac{\gamma h}{2(1-\gamma h)^2}\left\langle T_{g_k}\mathsf{L}_{g_k^{-1}}\nabla U(g_k), \log g_*^{-1}g_k \right\rangle \qquad\qquad III_1$$

$$- \frac{h^2}{4(1-\gamma h)^2}\|\nabla U(g_k)\|^2 \qquad\qquad III_3$$

Take a closer look:

$$\frac{1}{2}I_1 + III_2 = \frac{h}{2(1-\gamma h)}\left\langle T_{g_k}\mathsf{L}_{g_k^{-1}}\nabla U(g_k), \xi_k \right\rangle - \frac{h}{2(1-\gamma h)^2}\left\langle T_{g_k}\mathsf{L}_{g_k^{-1}}\nabla U(g_k), \xi_{k+1} \right\rangle$$

$$= \frac{h}{2(1-\gamma h)}\left\langle T_{g_k}\mathsf{L}_{g_k^{-1}}\nabla U(g_k), \xi_k - \frac{1}{1-\gamma h}\xi_{k+1} \right\rangle$$

$$= \frac{h^2}{2(1-\gamma h)^2}\left\| T_{g_k}\mathsf{L}_{g_k^{-1}}\nabla U(g_k)\right\|^2$$

$$\frac{1}{2}I_1 + II_2 + II_3 + III_3 = -\frac{1}{4}\left\| \xi_{k+1} - \xi_k + \frac{h}{1-\gamma h}T_{g_k}\mathsf{L}_{g_k^{-1}}\nabla U(g_k)\right\|^2 \le 0$$

Now we sum everything up

$$\mathcal{L}_{k+1}^{\mathrm{HB}} - \mathcal{L}_k^{\mathrm{HB}} \le -\frac{\gamma h}{2(1-\gamma h)}\left( \left\langle T_{g_k}\mathsf{L}_{g_k^{-1}}\nabla U(g_k), \log g_*^{-1}g_k \right\rangle + \|\xi_{k+1}\|^2 \right) + \frac{h^2}{2(1-\gamma h)}\|\nabla U(g_k)\|^2$$

By Eq. (4) (strong convexity) and Eq. (22) (corollary of co-coercivity) of $U$, we have $U(g_k) - U(g_*) + \frac{\mu}{2}\left\|\log g_*^{-1}g_k\right\|^2 \le \left\langle T_{g_k}\mathsf{L}_{g_k^{-1}}\nabla U(g_k), \log g_*^{-1}g_k \right\rangle$

$$\mathcal{L}_{k+1}^{\mathrm{HB}} - \mathcal{L}_k^{\mathrm{HB}} \le -\frac{\gamma h}{2(1-\gamma h)}\left( \frac{1}{4}(U(g_k) - U(g_*)) + \frac{\mu}{2}\left\|\log g_*^{-1}g_k\right\|^2 + \|\xi_{k+1}\|^2 \right)$$

$$- \frac{h}{2(1-\gamma h)}\left( \frac{3}{4}\gamma(U(g_k) - U(g_*)) - \frac{h}{1-\gamma h}\|\nabla U(g_k)\|^2 \right)$$

$$\le -\frac{\gamma h}{2(1-\gamma h)}\left( \frac{1}{4}(U(g_k) - U(g_*)) + \frac{\mu}{2}\left\|\log g_*^{-1}g_k\right\|^2 + \|\xi_{k+1}\|^2 \right) \qquad (31)$$

$$- \frac{h}{2(1-\gamma h)}\left( \frac{3\gamma}{4L} - \frac{h}{1-\gamma h} \right)\|\nabla U(g_k)\|^2$$

Cauchy-Schwarz inequality gives

$$\mathcal{L}_{k+1}^{\mathrm{HB}} \le \frac{1}{1-\gamma h}(U(g_k) - U(g_*)) + \frac{\gamma^2}{2(1-\gamma h)^2}\left\|\log g_*^{-1}g_k\right\|^2 + \frac{3}{4}\|\xi_{k+1}\|^2 \qquad (32)$$

Comparing the coefficients in Eq. (31) and (32) gives us the desired result. $\qquad\square$

**Lemma 32** (Monotonicity of the Lyapunov function for Heavy-Ball scheme)**.** *Assume the conditions in Lemma 31 is satisfied. By choosing* $\gamma = 2\sqrt{\mu}$ *and step size* $h = \frac{\sqrt{\mu}}{4L}$, *we have* $b_{HB} = \frac{\mu}{16L}$ *and*

$$\mathcal{L}_{k+1}^{HB} \le c_{HB}\mathcal{L}_k^{HB} \qquad (33)$$

*by defining* $c_{HB} := (1 + b_{HB})^{-1}$.

*Proof of Lemma 32.* By our choice of $h$ and $\gamma$ makes $\frac{3\gamma}{4L} - \frac{h}{1-\gamma h} \ge 0$, and the convergence rate by Lemma 31. $\qquad\square$

*Proof of Thm. 13.* This is a direct corollary of Lemma 32 and Cor. 12. $\qquad\square$

# D    More details about NAG-SC discretization

**Lemma 33.** *Assume* $\mathrm{ad}_X$ *is skew-adjoint for any* $X \in \mathfrak{g}$. *Suppose there is a geodesically convex set* $S \subset \mathsf{G}$ *satisfying:*

- *$U$ is geodesically $\mu$-strongly convex on a convex set $S \subset \mathsf{G}$.*

- *$g_*$ is the minima of $U$ on $S$.*

- *$g_k \in S$ for all $t \ge 0$.*

- $\log g_*^{-1} g$ and its differential is well-defined for all $g \in S$.

- $\max_{g \in S} d(g_*, g) \le \frac{a}{A}$ for some $a < 2\pi$.

Setting $h = \min\left\{\frac{1}{\sqrt{2L}}, \frac{1}{2p(a)}\right\}$ and $\gamma = 2\sqrt{\mu}$, we have

$$\mathcal{L}_{k+1}^{\text{NAG-SC}} - \mathcal{L}_k^{\text{NAG-SC}} \le b_{\text{NAG-SC}} \mathcal{L}_{k+1}^{\text{NAG-SC}}$$

for the contraction rate $b_{\text{NAG-SC}}$ given by

$$b_{\text{NAG-SC}} := \frac{1}{30} \sqrt{\mu} \min\left\{\frac{1}{\sqrt{2L}}, \frac{1}{2p(a)}\right\} \tag{34}$$

*Proof of Lemma 33.* For $(g_k, \xi_k)$ following the NAG-SC scheme Eq. (13), we define the shorthand notation $\mathcal{L}_k^{\text{NAG-SC}} := \mathcal{L}^{\text{NAG-SC}}(g_k, \xi_k)$, which gives

$$\mathcal{L}_k^{\text{NAG-SC}} = \frac{1}{1-\gamma h} \left(U(g_{k-1}) - U(g_*)\right) + \frac{1}{4}\|\xi_k\|^2 + \frac{1}{4}\left\|\xi_k + \frac{\gamma}{1-\gamma h}\log g_*^{-1} g_k + h\nabla U(g_{k-1})\right\|^2 - \frac{h^2(2-\gamma h)}{4(1-\gamma h)}\|\nabla U(g_{k-1})\|^2 \tag{35}$$

Evaluate of the basic terms of $\mathcal{L}_{k+1}^{\text{NAG-SC}} - \mathcal{L}_k^{\text{NAG-SC}}$ first:

•The first term: By the property of convex $L$-smooth functions in Eq. (21),

$$\frac{1}{1-\gamma h}\left(U(g_k) - U(g_{k-1})\right)$$

$$\le \frac{h}{1-\gamma h}\left\langle T_{g_k} \mathsf{L}_{g_k^{-1}} \nabla U(g_k), \xi_k \right\rangle \qquad\qquad I_1$$

$$- \frac{1}{2L}\frac{1}{1-\gamma h}\left\| T_{g_k}\mathsf{L}_{g_k^{-1}}\nabla U(g_k) - T_{g_{k-1}}\mathsf{L}_{g_{k-1}^{-1}}\nabla U(g_{k-1})\right\|^2 \qquad\qquad I_2$$

•The second term: NAG-SC scheme in Eq. 13 gives the following:

$$\frac{1}{4}\left(\|\xi_{k+1}\|^2 - \|\xi_k\|^2\right)$$

$$= \frac{1}{2}\langle \xi_{k+1} - \xi_k, \xi_{k+1}\rangle - \frac{1}{4}\|\xi_{k+1} - \xi_k\|^2$$

$$= -\frac{\gamma h}{2(1-\gamma h)}\|\xi_{k+1}\|^2 - \frac{h}{2}\left\langle T_{g_k}\mathsf{L}_{g_k^{-1}}\nabla U(g_k) - T_{g_{k-1}}\mathsf{L}_{g_{k-1}^{-1}}\nabla U(g_{k-1}), \xi_{k+1}\right\rangle$$

$$- \frac{h}{2(1-\gamma h)}\left\langle T_{g_{k-1}}\mathsf{L}_{g_{k-1}^{-1}}\nabla U(g_{k-1}), \xi_{k+1}\right\rangle - \frac{1}{4}\|\xi_{k+1} - \xi_k\|^2$$

$$= -\frac{\gamma h}{2(1-\gamma h)}\|\xi_{k+1}\|^2 \qquad\qquad II_1$$

$$- \frac{h(1-\gamma h)}{2}\left\langle T_{g_k}\mathsf{L}_{g_k^{-1}}\nabla U(g_k) - T_{g_{k-1}}\mathsf{L}_{g_{k-1}^{-1}}\nabla U(g_{k-1}), \xi_k\right\rangle \qquad\qquad II_2$$

$$+ \frac{h^2(1-\gamma h)}{2}\left\| T_{g_k}\mathsf{L}_{g_k^{-1}}\nabla U(g_k) - T_{g_{k-1}}\mathsf{L}_{g_{k-1}^{-1}}\nabla U(g_{k-1})\right\|^2 \qquad\qquad II_3$$

$$+ \frac{h^2}{2}\left\langle T_{g_k}\mathsf{L}_{g_k^{-1}}\nabla U(g_k) - T_{g_{k-1}}\mathsf{L}_{g_{k-1}^{-1}}\nabla U(g_{k-1}), T_{g_k}\mathsf{L}_{g_k^{-1}}\nabla U(g_k)\right\rangle \qquad\qquad II_4$$

$$- \frac{h}{2(1-\gamma h)}\left\langle \xi_{k+1}, T_{g_k}\mathsf{L}_{g_k^{-1}}\nabla U(g_k)\right\rangle \qquad\qquad II_5$$

$$- \frac{1}{4}\|\xi_{k+1} - \xi_k\|^2 \qquad\qquad II_6$$

•The third term: We consider the parametric curve on $\mathsf{G}$ connecting $g_k$ and $g_{k+1}$ defined by $g_t = g_k \exp(t\xi_{k+1}), t \in [0, h]$.

$$\frac{1}{4}\left\|\xi_{k+1} + \frac{\gamma}{1-\gamma h}\log g_*^{-1}g_{k+2} + hT_{g_k}\mathsf{L}_{g_k^{-1}}\nabla U(g_k)\right\|^2 - \frac{1}{4}\left\|\xi_k + \frac{\gamma}{1-\gamma h}\log g_*^{-1}g_k + hT_{g_{k-1}}\mathsf{L}_{g_{k-1}^{-1}}\nabla U(g_{k-1})\right\|^2$$

$$= \frac{1}{4}\left\|\xi_{k+1} + \frac{\gamma}{1-\gamma h}\log g_*^{-1}g_{k+2} + hT_{g_k}\mathsf{L}_{g_k^{-1}}\nabla U(g_k)\right\|^2 - \frac{1}{4}\left\|\xi_{k+1} + \frac{\gamma}{1-\gamma h}\log g_*^{-1}g_k + hT_{g_k}\mathsf{L}_{g_k^{-1}}\nabla U(g_k)\right\|^2$$

$$+ \frac{1}{4}\left\|\xi_{k+1} + \frac{\gamma}{1-\gamma h}\log g_*^{-1}g_k + hT_{g_k}\mathsf{L}_{g_k^{-1}}\nabla U(g_k)\right\|^2 - \frac{1}{4}\left\|\xi_k + \frac{\gamma}{1-\gamma h}\log g_*^{-1}g_k + hT_{g_{k-1}}\mathsf{L}_{g_{k-1}^{-1}}\nabla U(g_{k-1})\right\|^2$$

$$= \frac{1}{2}\int_0^h \left\langle \xi_{k+1} + \frac{\gamma}{1-\gamma h}\log g_*^{-1}g_t + hT_{g_k}\mathsf{L}_{g_k^{-1}}\nabla U(g_k), \frac{\gamma}{1-\gamma h}\,\mathrm{d}_{\xi_{k+1}}\log g_*^{-1}g_t\right\rangle \mathrm{d}t$$

$$+ \frac{1}{4}\left\langle \xi_{k+1} + \xi_k + \frac{2\gamma}{1-\gamma h}\log g_*^{-1}g_k + hT_{g_k}\mathsf{L}_{g_k^{-1}}\nabla U(g_k) + hT_{g_{k-1}}\mathsf{L}_{g_{k-1}^{-1}}\nabla U(g_{k-1}), \xi_{k+1} - \xi_k + hT_{g_k}\mathsf{L}_{g_k^{-1}}\nabla U(g_k) - hT_{g_{k-1}}\mathsf{L}_{g_{k-1}^{-1}}\nabla U(g_{k-1})\right\rangle$$

$$= \frac{1}{2}\int_0^h \left\langle \xi_{k+1} + \frac{\gamma}{1-\gamma h}\log g_*^{-1}g_t + hT_{g_k}\mathsf{L}_{g_k^{-1}}\nabla U(g_k), \frac{\gamma}{1-\gamma h}\,\mathrm{d}_{\xi_{k+1}}\log g_*^{-1}g_t\right\rangle \mathrm{d}t$$

$$+ \frac{1}{4(1-\gamma h)^2}\left\langle (2-\gamma h)\xi_{k+1} + 2\gamma \log g_*^{-1}g_k + (3-2\gamma h)hT_{g_k}\mathsf{L}_{g_k^{-1}}\nabla U(g_k), -\gamma h\xi_{k+1} - hT_{g_k}\mathsf{L}_{g_k^{-1}}\nabla U(g_k)\right\rangle$$

First, we estimate the term $\int_0^h \left\langle \xi_{k+1} + \frac{\gamma}{1-\gamma h}\log g_*^{-1}g_t + hT_{g_k}\mathsf{L}_{g_k^{-1}}\nabla U(g_k), \frac{\gamma}{1-\gamma h}\,\mathrm{d}_{\xi_{k+1}}\log g_*^{-1}g_t\right\rangle \mathrm{d}t$ using the property of $\mathrm{d}\log$ in Cor. 16, 17 and 18.

$$\left\langle \frac{\gamma}{1-\gamma h}\log g_*^{-1}g_t + \xi_{k+1} + hT_{g_k}\mathsf{L}_{g_k^{-1}}\nabla U(g_k), \mathrm{d}_{\xi_{k+1}}\log g_*^{-1}g_t\right\rangle$$

$$= \left\langle \frac{\gamma}{1-\gamma h}\log g_*^{-1}g_t + \xi_{k+1}, \mathrm{d}_{\xi_{k+1}}\log g_*^{-1}g_t\right\rangle$$

$$+ h\left\langle T_{g_k}\mathsf{L}_{g_k^{-1}}\nabla U(g_k), \xi_{k+1}\right\rangle + h\left\langle T_{g_k}\mathsf{L}_{g_k^{-1}}\nabla U(g_k), \mathrm{d}_{\xi_{k+1}}\log g_*^{-1}g_t - \xi_{k+1}\right\rangle$$

$$\le \frac{\gamma}{1-\gamma h}\left\langle \log g_*^{-1}g_t, \xi_{k+1}\right\rangle + \|\xi_{k+1}\|^2 + h\left\langle T_{g_k}\mathsf{L}_{g_k^{-1}}\nabla U(g_k), \xi_{k+1}\right\rangle + p(a)h\|\nabla U(g_k)\|\|\xi_{k+1}\| \quad (36)$$

Taking integral gives

$$\int_0^h \left\langle \frac{\gamma}{1-\gamma h}\log g_*^{-1}g_t + \xi_{k+1} + hT_{g_k}\mathsf{L}_{g_k^{-1}}\nabla U(g_k), \mathrm{d}_{\xi_{k+1}}\log g_*^{-1}g_t\right\rangle \mathrm{d}t$$

$$\le \int_0^h \frac{\gamma}{1-\gamma h}\left\langle \log g_*^{-1}g_t, \xi_{k+1}\right\rangle + \|\xi_{k+1}\|^2 + h\left\langle T_{g_k}\mathsf{L}_{g_k^{-1}}\nabla U(g_k), \xi_{k+1}\right\rangle + p(a)h\|\nabla U(g_k)\|\|\xi_{k+1}\| \,\mathrm{d}t$$

$$= \frac{\gamma}{1-\gamma h}\int_0^h \left\langle \log g_*^{-1}g_t, \xi_{k+1}\right\rangle \mathrm{d}t + h\|\xi_{k+1}\|^2 + h^2\left\langle T_{g_k}\mathsf{L}_{g_k^{-1}}\nabla U(g_k), \xi_{k+1}\right\rangle + h^2 p(a)\|\nabla U(g_k)\|\|\xi_{k+1}\|$$

We calculate the integral

$$\int_0^h \left\langle \log g_*^{-1}g_t, \xi_{k+1}\right\rangle \mathrm{d}t$$

$$= \int_0^h \left[\left\langle \log g_*^{-1}g_k, \xi_{k+1}\right\rangle + \int_0^t \left\langle \mathrm{d}_{\xi_{k+1}}\log g_*^{-1}g_\tau, \xi_{k+1}\right\rangle \mathrm{d}\tau\right]\mathrm{d}t$$

$$\le h\left\langle \log g_*^{-1}g_k, \xi_{k+1}\right\rangle + \int_0^h\int_0^t \|\xi_{k+1}\|^2 \,\mathrm{d}\tau\,\mathrm{d}t$$

$$= h\left\langle \log g_*^{-1}g_k, \xi_{k+1}\right\rangle + \frac{h^2}{2}\|\xi_{k+1}\|^2$$

which gives us

$$\int_0^h \left\langle \xi_{k+1} + \frac{\gamma}{1-\gamma h}\log g_*^{-1}g_t + hT_{g_k}\mathsf{L}_{g_k^{-1}}\nabla U(g_k), \mathrm{d}_{\xi_{k+1}}\log g_*^{-1}g_t\right\rangle \mathrm{d}t$$

$$\le \frac{\gamma h}{1-\gamma h}\left\langle \log g_*^{-1}g_k, \xi_{k+1}\right\rangle + \frac{h(2-\gamma h)}{2(1-\gamma h)}\|\xi_{k+1}\|^2$$

$$+ h^2\left\langle T_{g_k}\mathsf{L}_{g_k^{-1}}\nabla U(g_k), \xi_{k+1}\right\rangle + h^2 p(a)\|\nabla U(g_k)\|\|\xi_{k+1}\|$$

The other term

$$\left\langle (2-\gamma h)\xi_{k+1} + 2\gamma \log g_*^{-1}g_k + (3-2\gamma h)hT_{g_k}\mathsf{L}_{g_k^{-1}}\nabla U(g_k), -\gamma h\xi_{k+1} - hT_{g_k}\mathsf{L}_{g_k^{-1}}\nabla U(g_k)\right\rangle$$

$$= -\gamma h(2-\gamma h)\|\xi_{k+1}\|^2 - 2\gamma^2 h\left\langle \log g_*^{-1}g_k, \xi_{k+1}\right\rangle - \gamma h^2(3-2\gamma h)\left\langle T_{g_k}\mathsf{L}_{g_k^{-1}}\nabla U(g_k), \xi_{k+1}\right\rangle$$

$$- h(2-\gamma h)\left\langle T_{g_k}\mathsf{L}_{g_k^{-1}}\nabla U(g_k), \xi_{k+1}\right\rangle - 2\gamma h\left\langle \log g_*^{-1}g_k, T_{g_k}\mathsf{L}_{g_k^{-1}}\nabla U(g_k)\right\rangle - h^2(3-2\gamma h)\left\|T_{g_k}\mathsf{L}_{g_k^{-1}}\nabla U(g_k)\right\|^2$$

$$= -\gamma h(2-\gamma h)\|\xi_{k+1}\|^2 - 2\gamma^2 h\left\langle \log g_*^{-1}g_k, \xi_{k+1}\right\rangle - 2h(1+\gamma h - \gamma^2 h^2)\left\langle T_{g_k}\mathsf{L}_{g_k^{-1}}\nabla U(g_k), \xi_{k+1}\right\rangle$$

$$- 2\gamma h\left\langle \log g_*^{-1}g_k, T_{g_k}\mathsf{L}_{g_k^{-1}}\nabla U(g_k)\right\rangle - h^2(3-2\gamma h)\|\nabla U(g_k)\|^2$$

Summing them up, we have

$$\left\|\xi_{k+1} + \frac{\gamma}{1-\gamma h}\log g_*^{-1}g_{k+1} + hT_{g_k}\mathsf{L}_{g_k^{-1}}\nabla U(g_k)\right\|^2 - \left\|\xi_k + \frac{\gamma}{1-\gamma h}\log g_*^{-1}g_k + hT_{g_{k-1}}\mathsf{L}_{g_{k-1}^{-1}}\nabla U(g_{k-1})\right\|^2$$

$$\leq \frac{2\gamma^2 h}{(1-\gamma h)^2}\left\langle \log g_*^{-1}g_k, \xi_{k+1}\right\rangle + \frac{\gamma h(2-\gamma h)}{(1-\gamma h)^2}\|\xi_{k+1}\|^2$$

$$+ \frac{2\gamma h^2}{1-\gamma h}\left\langle T_{g_k}\mathsf{L}_{g_k^{-1}}\nabla U(g_k), \xi_{k+1}\right\rangle + \frac{2\gamma h^2}{1-\gamma h}p(a)\|\nabla U(g_k)\|\|\xi_{k+1}\|$$

$$- \frac{\gamma h(2-\gamma h)}{(1-\gamma h)^2}\|\xi_{k+1}\|^2 - \frac{2\gamma^2 h}{(1-\gamma h)^2}\left\langle \log g_*^{-1}g_k, \xi_{k+1}\right\rangle - \frac{2h(1+\gamma h - \gamma^2 h^2)}{(1-\gamma h)^2}\left\langle T_{g_k}\mathsf{L}_{g_k^{-1}}\nabla U(g_k), \xi_{k+1}\right\rangle$$

$$- \frac{2\gamma h}{(1-\gamma h)^2}\left\langle \log g_*^{-1}g_k, T_{g_k}\mathsf{L}_{g_k^{-1}}\nabla U(g_k)\right\rangle - \frac{h^2(3-2\gamma h)}{(1-\gamma h)^2}\left\|T_{g_k}\mathsf{L}_{g_k^{-1}}\nabla U(g_k)\right\|^2$$

$$\leq \frac{2\gamma h^2}{1-\gamma h}p(a)\|\nabla U(g_k)\|\|\xi_{k+1}\| - \frac{2h}{(1-\gamma h)^2}\left\langle T_{g_k}\mathsf{L}_{g_k^{-1}}\nabla U(g_k), \xi_{k+1}\right\rangle$$

$$- \frac{2\gamma h}{(1-\gamma h)^2}\left\langle \log g_*^{-1}g_k, T_{g_k}\mathsf{L}_{g_k^{-1}}\nabla U(g_k)\right\rangle - \frac{h^2(3-2\gamma h)}{(1-\gamma h)^2}\|\nabla U(g_k)\|^2$$

Eventually, the third term becomes

$$\frac{1}{4}\left\|\xi_{k+1} + \frac{\gamma}{1-\gamma h}\log g_*^{-1}g_{k+1} + hT_{g_k}\mathsf{L}_{g_k^{-1}}\nabla U(g_k)\right\|^2 - \frac{1}{4}\left\|\xi_k + \frac{\gamma}{1-\gamma h}\log g_*^{-1}g_k + hT_{g_{k-1}}\mathsf{L}_{g_{k-1}^{-1}}\nabla U(g_{k-1})\right\|^2$$

$$= -\frac{h\gamma}{2(1-\gamma h)^2}\left\langle T_{g_k}\mathsf{L}_{g_k^{-1}}\nabla U(g_k), \log g_*^{-1}g_k\right\rangle \qquad\qquad III_1$$

$$- \frac{h}{2(1-\gamma h)^2}\left\langle T_{g_k}\mathsf{L}_{g_k^{-1}}\nabla U(g_k), \xi_{k+1}\right\rangle \qquad\qquad III_2$$

$$- \frac{h^2}{2(1-\gamma h)}\left\|T_{g_k}\mathsf{L}_{g_k^{-1}}\nabla U(g_k)\right\|^2 \qquad\qquad III_3$$

$$- \frac{h^2}{4(1-\gamma h)^2}\left\|T_{g_k}\mathsf{L}_{g_k^{-1}}\nabla U(g_k)\right\|^2 \qquad\qquad III_4$$

$$+ \frac{\gamma h^2}{2(1-\gamma h)}p(a)\|\nabla U(g_k)\|\|\xi_{k+1}\| \qquad\qquad \text{extra term from curvature}$$

We sum the all the terms up to get

$$\mathcal{L}_{k+1}^{\text{NAG-SC}} - \mathcal{L}_{k}^{\text{NAG-SC}}$$

$$\leq -\frac{\gamma h}{2(1-\gamma h)}\left(\frac{1}{1-\gamma h}\left\langle T_{g_k}\mathsf{L}_{g_k^{-1}}\nabla U(g_k), \log g_*^{-1}g_k\right\rangle + \|\xi_{k+1}\|^2\right) \qquad II_1 + III_1$$

$$-\frac{h}{2(1-\gamma h)}\left\langle T_{g_k}\mathsf{L}_{g_k^{-1}}\nabla U(g_k), \frac{1}{1-\gamma h}\xi_{k+1} - \xi_k + hT_{g_k}\mathsf{L}_{g_k^{-1}}\nabla U(g_k)\right\rangle \qquad \tfrac{1}{2}I_1 + III_2 + III_3$$

$$-\frac{h(1-\gamma h)}{2}\left\langle T_{g_k}\mathsf{L}_{g_k^{-1}}\nabla U(g_k) - T_{g_{k-1}}\mathsf{L}_{g_{k-1}^{-1}}\nabla U(g_{k-1}), \xi_k\right\rangle \qquad II_2$$

$$+\frac{h^2}{2}\left\langle T_{g_k}\mathsf{L}_{g_k^{-1}}\nabla U(g_k) - T_{g_{k-1}}\mathsf{L}_{g_{k-1}^{-1}}\nabla U(g_{k-1}), T_{g_k}\mathsf{L}_{g_k^{-1}}\nabla U(g_k)\right\rangle \qquad II_4$$

$$-\frac{1}{4}\left(\frac{2h}{1-\gamma h}\left\langle\xi_{k+1}-\xi_k, T_{g_k}\mathsf{L}_{g_k^{-1}}\nabla U(g_k)\right\rangle + \|\xi_{k+1}-\xi_k\|^2 + \frac{h^2}{(1-\gamma h)^2}\|\nabla U(g_k)\|^2\right)$$
$$\qquad \tfrac{1}{2}I_1 + II_5 + II_6 + III_4$$

$$-\frac{1}{2}\left(\frac{1}{L}\frac{1}{1-\gamma h} - h^2(1-\gamma h)\right)\left\|T_{g_k}\mathsf{L}_{g_k^{-1}}L_\nabla U(g_k) - T_{g_{k-1}}\mathsf{L}_{g_{k-1}^{-1}}\nabla U(g_{k-1})\right\|^2 \qquad I_2 + II_3$$

$$-\frac{h^2(2-\gamma h)}{4(1-\gamma h)}\left(\|\nabla U(g_k)\|^2 - \|\nabla U(g_{k-1})\|^2\right) \qquad \text{Additional term}$$

$$+\frac{\gamma h^2}{2(1-\gamma h)}p(a)\|\nabla U(g_k)\|\|\xi_{k+1}\| \qquad \text{extra term from curvature}$$

$$\frac{1}{2}I_1 + III_2 + III_3$$

$$= \frac{h^2}{2(1-\gamma h)}\left\langle T_{g_k}\mathsf{L}_{g_k^{-1}}\nabla U(g_k) - T_{g_{k-1}}\mathsf{L}_{g_{k-1}^{-1}}\nabla U(g_{k-1}), T_{g_k}\mathsf{L}_{g_k^{-1}}\nabla U(g_k)\right\rangle \qquad IV_1$$

$$+\frac{\gamma h^3}{2(1-\gamma h)^2}\left\|T_{g_k}\mathsf{L}_{g_k^{-1}}\nabla U(g_k)\right\|^2 \qquad IV_2$$

$$\mathcal{L}_{k+1}^{\text{NAG-SC}} - \mathcal{L}_{k}^{\text{NAG-SC}}$$

$$\leq \frac{\gamma h}{2(1-\gamma h)}\left(\frac{1}{1-\gamma h}\left(\left\langle T_{g_k}\mathsf{L}_{g_k^{-1}}\nabla U(g_k), \log g_*^{-1}g_k\right\rangle - h^2\|\nabla U(g_k)\|^2\right) + \|\xi_{k+1}\|^2\right)$$
$$\qquad II_1 + III_1 + IV_2$$

$$-\frac{1-\gamma h}{2}\left\langle T_{g_k}\mathsf{L}_{g_k^{-1}}\nabla U(g_k) - T_{g_{k-1}}\mathsf{L}_{g_{k-1}^{-1}}\nabla U(g_{k-1}), \log g_{k-1}^{-1}g_k\right\rangle \qquad II_2$$

$$+\frac{h^2}{2}\frac{2-\gamma h}{1-\gamma h}\left\langle T_{g_k}\mathsf{L}_{g_k^{-1}}\nabla U(g_k) - T_{g_{k-1}}\mathsf{L}_{g_{k-1}^{-1}}\nabla U(g_{k-1}), T_{g_{k-1}}\mathsf{L}_{g_{k-1}^{-1}}\nabla U(g_{k-1})\right\rangle \qquad II_4 + IV_1$$

$$-\frac{1}{2}\left(\frac{1}{L}\frac{1}{1-\gamma h} - h^2(1-\gamma h)\right)\left\|T_{g_k}\mathsf{L}_{g_k^{-1}}\nabla U(g_k) - T_{g_{k-1}}\mathsf{L}_{g_{k-1}^{-1}}\nabla U(g_{k-1})\right\|^2 \qquad I_2 + II_3$$

$$-\frac{h^2}{4}\frac{2-\gamma h}{1-\gamma h}\left(\|\nabla U(g_k)\|^2 - \|\nabla U(g_{k-1})\|^2\right) \qquad \text{Additional term}$$

$$+\frac{\gamma h^2}{2(1-\gamma h)}p(a)\|\nabla U(g_k)\|\|\xi_{k+1}\| \qquad \text{extra term from curvature}$$

$$II_4 + IV_1 + \text{Additional term} = \frac{h^2}{4}\frac{2-\gamma h}{1-\gamma h}\left\|T_{g_k}\mathsf{L}_{g_k^{-1}}\nabla U(g_k) - T_{g_{k-1}}\mathsf{L}_{g_{k-1}^{-1}}\nabla U(g_{k-1})\right\|^2$$

$$(II_4 + IV_1 + \text{Additional term}) + (I_2 + II_3)$$

$$\leq \frac{h^2}{2}\left(1 - \gamma h + \frac{1}{1-\gamma h} - \frac{1}{1-\gamma h}\frac{1}{Lh^2}\right)\left\|T_{g_k}\mathsf{L}_{g_k^{-1}}\nabla U(g_k) - T_{g_{k-1}}\mathsf{L}_{g_{k-1}^{-1}}\nabla U(g_{k-1})\right\|^2$$

$$\mathcal{L}_{k+1}^{\text{NAG-SC}} - \mathcal{L}_k^{\text{NAG-SC}}$$

$$\leq -\frac{\gamma h}{2(1-\gamma h)}\left(\frac{1}{1-\gamma h}\left(\left\langle T_{g_k}\mathsf{L}_{g_k^{-1}}\nabla U(g_k), \log g_*^{-1}g_k\right\rangle - h^2\|\nabla U(g_k)\|^2\right) + \|\xi_{k+1}\|\right)$$

$$- \frac{1-\gamma h}{2}\left\langle T_{g_k}\mathsf{L}_{g_k^{-1}}\nabla U(g_k) - T_{g_{k-1}}\mathsf{L}_{g_{k-1}^{-1}}\nabla U(g_{k-1}), \log g_{k-1}^{-1}g_k\right\rangle$$

$$+ \frac{h^2}{2}\left(1-\gamma h + \frac{1}{1-\gamma h} - \frac{1}{1-\gamma h}\frac{1}{Lh^2}\right)\left\|T_{g_k}\mathsf{L}_{g_k^{-1}}\nabla U(g_k) - T_{g_{k-1}}\mathsf{L}_{g_{k-1}^{-1}}\nabla U(g_{k-1})\right\|^2$$

$$+ \frac{\gamma h^2}{2(1-\gamma h)}p(a)\left\|T_{g_k}\mathsf{L}_{g_k^{-1}}\nabla U(g_k)\right\|\|\xi_{k+1}\| \qquad\qquad \text{extra term from curvature}$$

By the property of $L$-smoothness in Eq. (22), we have

$$\left\langle T_{g_k}\mathsf{L}_{g_k^{-1}}\nabla U(g_k) - T_{g_{k-1}}\mathsf{L}_{g_{k-1}^{-1}}\nabla U(g_{k-1}), \log g_{k-1}^{-1}g_k\right\rangle \geq \frac{1}{L}\left\|T_{g_k}\mathsf{L}_{g_k^{-1}}\nabla U(g_k) - T_{g_{k-1}}\mathsf{L}_{g_{k-1}^{-1}}\nabla U(g_{k-1})\right\|^2$$

and

$$\mathcal{L}_{k+1}^{\text{NAG-SC}} - \mathcal{L}_k^{\text{NAG-SC}} \leq -\frac{\gamma h}{2(1-\gamma h)}\left(\frac{1}{1-\gamma h}\left(\left\langle T_{g_k}\mathsf{L}_{g_k^{-1}}\nabla U(g_k), \log g_*^{-1}g_k\right\rangle - h^2\|\nabla U(g_k)\|^2\right) + \|\xi_{k+1}\|^2\right)$$

$$- \frac{1}{2}\left(1-\gamma h + \frac{1}{1-\gamma h}\right)\left(\frac{1}{L} - h^2\right)\left\|T_{g_k}\mathsf{L}_{g_k^{-1}}\nabla U(g_k) - T_{g_{k-1}}\mathsf{L}_{g_{k-1}^{-1}}\nabla U(g_{k-1})\right\|^2$$

$$+ \frac{\gamma h^2}{2(1-\gamma h)}p(a)\|\nabla U(g_k)\|\|\xi_{k+1}\|$$

When $h \leq \frac{1}{\sqrt{2L}}$, together with Lemma 33 and Cauchy-Schwarz inequality, we have

$$\mathcal{L}_{k+1}^{\text{NAG-SC}} - \mathcal{L}_k^{\text{NAG-SC}}$$

$$\leq -\frac{\gamma h}{2(1-\gamma h)}\left(\frac{1}{1-\gamma h}\left(\left\langle T_{g_k}\mathsf{L}_{g_k^{-1}}\nabla U(g_k), \log g_*^{-1}g_k\right\rangle - h^2\|\nabla U(g_k)\|^2\right) + \|\xi_{k+1}\|^2\right)$$

$$+ \frac{\gamma h^2}{2(1-\gamma h)}p(a)\|\nabla U(g_k)\|\|\xi_{k+1}\|$$

$$= -\frac{\gamma h}{2(1-\gamma h)}\left(\frac{1}{1-\gamma h}\left(\left\langle T_{g_k}\mathsf{L}_{g_k^{-1}}\nabla U(g_k), \log g_*^{-1}g_k\right\rangle - h^2\|\nabla U(g_k)\|^2\right) + \|\xi_{k+1}\|^2\right)$$

$$+ \frac{\gamma h^2}{4(1-\gamma h)}p(a)\left(\lambda\|\nabla U(g_k)\|^2 + \lambda^{-1}\|\xi_{k+1}\|^2\right)$$

where $\lambda > 0$ is a parameter to be chosen later. Using property of $L$-smoothness in Eq. (21), (22) and property of $\mu$-strong convexity in Eq. (24), we have for any $\lambda_1 \geq 0$ and $\lambda_2 \geq 1$ satisfying $\lambda_1 + \lambda_2 \geq 1$,

$$\frac{2(1-\gamma h)^2}{\gamma h}\left(\mathcal{L}_{k+1}^{\text{NAG-SC}} - \mathcal{L}_k^{\text{NAG-SC}}\right)$$

$$\leq -\left\langle T_{g_k}\mathsf{L}_{g_k^{-1}}\nabla U(g_k), \log g_*^{-1}g_k\right\rangle$$

$$+ h^2\|\nabla U(g_k)\|^2 - (1-\gamma h)\|\xi_{k+1}\|^2 + \frac{h(1-\gamma h)\lambda p(a)}{2}\|\nabla U(g_k)\|^2 + \frac{h(1-\gamma h)\lambda^{-1}p(a)}{2}\|\xi_{k+1}\|^2$$

$$\leq -(2-\lambda_1-\lambda_2)(U(g_k) - U(g_*)) - \frac{\lambda_1}{2L}\|\nabla U(g_k)\|^2 - \frac{\mu\lambda_2}{2}d^2(g,g_*)$$

$$+ h^2\|\nabla U(g_k)\|^2 - (1-\gamma h)\|\xi_{k+1}\|^2 + \frac{h(1-\gamma h)\lambda p(a)}{2}\|\nabla U(g_k)\|^2 + \frac{h(1-\gamma h)\lambda^{-1}p(a)}{2}\|\xi_{k+1}\|^2$$

$$\leq -(2-\lambda_1-\lambda_2)(U(g_k) - U(g_*)) + \left(\frac{h(1-\gamma h)\lambda p(a)}{2} + h^2 - \frac{\lambda_1}{2L}\right)\|\nabla U(g_k)\|^2 - \frac{\mu\lambda_2}{2}d^2(g,g_*)$$

$$- (1-\gamma h)\left(1 - \frac{h\lambda^{-1}p(a)}{2}\right)\|\xi_{k+1}\|^2 \tag{37}$$

Now we try to upper bound $\mathcal{L}_k^{\text{NAG-SC}}$. Using Cauchy-Schwarz inequality,

$$\left\| \xi_k + \frac{2\gamma}{1-\gamma h} \log g_*^{-1} g_k + h T_{g_{k-1}} \mathsf{L}_{g_{k-1}^{-1}} \nabla U(g_{k-1}) \right\|^2$$

$$\leq 3 \left[ \|\xi_k\|^2 + \left(\frac{2\gamma}{1-\gamma h}\right)^2 d^2(g_*, g_k) + h^2 \|\nabla U(g_{k-1})\|^2 \right]$$

$$\leq 3 \left[ \|\xi_k\|^2 + \left(\frac{2\gamma}{1-\gamma h}\right)^2 (d(g_*, g_k) + h\|\xi_k\|)^2 + h^2 \|\nabla U(g_{k-1})\|^2 \right]$$

$$\leq 3 \left[ \left(1 + 2\left(\frac{2\gamma h}{1-\gamma h}\right)^2\right) \|\xi_k\|^2 + 2\left(\frac{2\gamma}{1-\gamma h}\right)^2 d^2(g_*, g_k) + h^2 \|\nabla U(g_{k-1})\|^2 \right]$$

As a result,

$\mathcal{L}_k^{\text{NAG-SC}}$

$$\leq \frac{1}{1-\gamma h} (U(g_{k-1}) - U(g_*)) + \left(\frac{1}{4} + \frac{3}{4} + \frac{6\gamma^2 h^2}{(1-\gamma h)^2}\right) \|\xi_k\|^2$$

$$+ \frac{6\gamma^2}{(1-\gamma h)^2} d^2(g_*, g_k) + \left(\frac{3}{4} - \frac{(2-\gamma h)}{4(1-\gamma h)}\right) h^2 \|\nabla U(g_{k-1})\|^2$$

$$= \frac{1}{1-\gamma h} (U(g_{k-1}) - U(g_*)) + \frac{1-2\gamma h + 7\gamma^2 h^2}{(1-\gamma h)^2} \|\xi_k\|^2 + \frac{6\gamma^2}{(1-\gamma h)^2} d^2(g_*, g_k) + \frac{1-2\gamma h}{4(1-\gamma h)} h^2 \|\nabla U(g_{k-1})\|^2$$

$$\leq \left(\frac{1}{1-\gamma h} + \frac{1-2\gamma h}{4(1-\gamma h)} \frac{h^2}{2L}\right) (U(g_{k-1}) - U(g_*)) + \frac{1-2\gamma h + 7\gamma^2 h^2}{4(1-\gamma h)^2} \|\xi_k\|^2 + \frac{3\gamma^2}{2(1-\gamma h)^2} d^2(g_*, g_k)$$

$$\leq \frac{5-2\gamma h}{4(1-\gamma h)} (U(g_{k-1}) - U(g_*)) + \frac{1-2\gamma h + 7\gamma^2 h^2}{(1-\gamma h)^2} \|\xi_k\|^2 + \frac{3\gamma^2}{2(1-\gamma h)^2} d^2(g_*, g_k)$$

which is the same as

$$2(1-\gamma h)\mathcal{L}_k^{\text{NAG-SC}}$$

$$\leq \frac{5-2\gamma h}{2} (U(g_{k-1}) - U(g_*)) + \frac{2(1-2\gamma h + 7\gamma^2 h^2)}{1-\gamma h} \|\xi_k\|^2 + \frac{3\gamma^2}{1-\gamma h} d^2(g_*, g_k) \qquad (38)$$

We suppose

$$\frac{h(1-\gamma h)\lambda p(a)}{2} + h^2 - \frac{\lambda_1}{2L} \leq 0 \qquad (39)$$

As a result, by comparing the parameters in Eq. (38) and (37), we have the contraction rate is

$$b_{\text{NAG-SC}} \geq \frac{\gamma h}{1-\gamma h} \min\left\{ \frac{2(2-\lambda_1-\lambda_2)}{5-2\gamma h}, \left(1 - \frac{h\lambda^{-1}p(a)}{2}\right) \frac{(1-\gamma h)^2}{2(1-2\gamma h+7\gamma^2 h^2)}, \frac{\mu\lambda_2(1-\gamma h)}{6\gamma^2} \right\}$$

$$= \gamma h \min\left\{ \frac{2(2-\lambda_1-\lambda_2)}{(5-2\gamma h)(1-\gamma h)}, \left(1 - \frac{h\lambda^{-1}p(a)}{2}\right) \frac{1-\gamma h}{2(1-2\gamma h+7\gamma^2 h^2)}, \frac{\mu\lambda_2}{6\gamma^2} \right\}$$

Now we try to give a lower bound for the contraction rate $c$ upon a set of parameters $(\lambda, \lambda_1, \lambda_2)$. By assuming $\gamma h \leq \frac{1}{7}$, we have

$$b_{\text{NAG-SC}} \geq \gamma h \min\left\{ 2(2-\lambda_1-\lambda_2), \frac{1}{2}\left(1 - \frac{h\lambda^{-1}p(a)}{2}\right), \frac{\mu\lambda_2}{6\gamma^2} \right\}$$

Choosing

$$\lambda_1 = 2L\left(\frac{h\lambda p(a)}{2} + h^2\right)$$

to make Eq. (39) satisfied. By assuming $h \leq \frac{1}{2L}$, we choose

$$\lambda_2 = \frac{1 - h\lambda p(a)}{1 + \frac{\mu}{12\gamma^2}}$$

to make $2(2 - \lambda_1 - \lambda_2) \geq \frac{\mu \lambda_2}{6\gamma^2}$. Then we have

$$b_{\text{NAG-SC}} \geq \gamma h \min \left\{ \frac{1}{2} \left( 1 - \frac{h\lambda^{-1}p(a)}{2} \right), \frac{\mu}{6(\gamma^2 + \mu)} \left( 1 - h\lambda p(a) \right) \right\}$$

$$\geq 2\sqrt{\mu} h \min \left\{ \frac{1}{2} \left( 1 - \frac{h\lambda^{-1}p(a)}{2} \right), \frac{1}{30} \left( 1 - h\lambda p(a) \right) \right\}$$

$$\geq 2\sqrt{\mu} h \min \left\{ \frac{1}{2} \left( 1 - \frac{hp(a)}{2} \right), \frac{1}{30} \left( 1 - hp(a) \right) \right\}$$

by choosing $\gamma = 2\sqrt{\mu}$, same as continuous case and simply choose $\lambda = 1$.

$$b_{\text{NAG-SC}} \geq 2\sqrt{\mu} \min \left\{ \frac{1}{4} h(2 - hp(a)), \frac{1}{30} h(1 - hp(a)) \right\}$$

Setting $h = \min \left\{ \frac{1}{\sqrt{2L}}, \frac{1}{2p(a)} \right\}$, we have

$$b_{\text{NAG-SC}} \geq 2\sqrt{\mu} \frac{h}{60}$$

$$= \frac{1}{30} \sqrt{\mu} \min \left\{ \frac{1}{\sqrt{2L}}, \frac{1}{2p(a)} \right\}$$

which gives us the desired result. $\qquad\square$

**Corollary 34.** *If the iteration for NAG-SC is initialized by $g_0 \in S_0$, with $(g_0, \xi_0)$ satisfying*

$$E^{NAG\text{-}SC}(g_0, \xi_0) \leq (1 - \gamma h)u$$

*for some $u$, with the $u$ sub-level set of $U$ satisfying Eq. (15). Then we have $g_k \in S_0$ for any $k$ for the NAG-SC scheme Eq. (13) when $h \leq \sqrt{\frac{1}{2L}}$.*

*Proof of Cor. 34.* First we prove $\mathcal{L}^{\text{NAG-SC}} \geq 1/4 \|\xi_k\|^2$. By $L$-smoothness and geodesic convexity, Lemma 23 gives

$$U(g) - U(g_*) \geq \frac{1}{L} \|\nabla U(g)\|^2 \quad \forall g \in S$$

As a result, when $h \leq \sqrt{\frac{1}{2L}}$, we have $\frac{1}{L(1 - \gamma h)} \geq \frac{h^2(2 - \gamma h)}{2(1 - \gamma h)}$, leading to $\mathcal{L}^{\text{NAG-SC}}(g, \xi) \geq \frac{1}{4} \|\xi\|^2$.

Now we are ready to prove this corollary by induction. Suppose we have $\mathfrak{g}_{k-1} \in S_0$. By the monotonicity of the Lyapunov function, $\mathcal{L}^{\text{NAG-SC}}(g_k, \xi_k) \leq \mathcal{L}^{\text{NAG-SC}}(g_0, \xi_0)$. As a result,

$$\|\xi_k\| \leq \sqrt{4\mathcal{L}_k^{\text{NAG-SC}}} \leq 2\sqrt{\mathcal{L}_0^{\text{NAG-SC}}}$$

Since we have $g_k = g_{k-1} \exp(h\xi_k)$, we have $d(g_k, g_{k-1}) \leq h\|\xi_k\|$. Together with the condition that $U(g_{k-1}) \leq (1 - \gamma h)\mathcal{L}_k^{\text{NAG-SC}} \leq (1 - \gamma h)u$, we have $g_k \in S_0$. Mathematical induction gives the desired result. $\qquad\square$

*Proof of Thm. 14.* This is the direct corollary from Lemma 33 and Cor. 34 when defining $c_{\text{NAG-SC}} := (1 + b_{\text{NAG-SC}})^{-1}$, with $b_{\text{NAG-SC}}$ is given in Eq. (34). $\qquad\square$

**Remark 35** (Why Lie NAG-SC losses convergence rate comparing to the Euclidean case)**.** *In order to utilize the extra term $h \left( T_{g_k} \mathsf{L}_{g_k^{-1}} \nabla U(g_k) - T_{g_{k-1}} \mathsf{L}_{g_{k-1}^{-1}} \nabla U(g_{k-1}) \right)$ in Eq. (13), Lyapunov function Eq. (14) has the term $\xi_k + \frac{\gamma}{1 - \gamma h} \log g_*^{-1} g_{k+1} + h \nabla U(g_k)$, and the term $\left\langle \log g_*^{-1} g_{k+1} - \log g_*^{-1} g_k, T_{g_k} \mathsf{L}_{g_k^{-1}} \nabla U(g_k) \right\rangle$ needs to be quantified, consequently. However, due to the non-linearity of the Lie group, we have to make the assumption that $g_{k+1}$ and $g_k$ are close to $g_*$, leading to the space is 'nearly linear', so that we can bound the error from the non-linearity of the space using Cor. 18. Please see the proof of Lemma 33, Eq. (36) for more details.*

*However, such loss of convergence rate due to curved spaces is also observed in some of the best results so far [33, 1].*

*For heavy-ball scheme, the design of the Lyapunov function only has the term $\frac{\gamma}{1 - \gamma h} \log g_*^{-1} g_k + \xi_k$, and we can use the properties Eq. (16) and (17) to quantify $\mathcal{L}_{k+1}^{HB} - \mathcal{L}_k^{HB}$.*

## D.1 Modified energy for NAG-SC

**Remark 36** (Why we cannot have an modified energy for NAG-SC). *An 'modified energy' for Lie NAG-SC is provided in Eq. (41), whose monotonicity is shown in Thm. 37. The modified energy is global defined and required only L-smoothness. However, the failure for this 'energy function' is because its monotonicity requires step size $\mathcal{O}\left(\frac{\sqrt{\mu}}{L}\right)$, which is smaller than the step size $\mathcal{O}\left(\frac{1}{\sqrt{L}}\right)$ that provides acceleration.*

*Comparing with the Heavy-Ball scheme, the larger step size and the acceleration of NAG-SC come from extra term $(1 - \gamma h)h\left(T_{g_k}\mathsf{L}_{g_k^{-1}}\nabla U(g_k) - T_{g_{k-1}}\mathsf{L}_{g_{k-1}^{-1}}\nabla U(g_{k-1})\right)$, which is closely related to co-coercivity in Lemma 23. However, co-coercivity requires (local) geodesic convexity and L-smoothness at the same time, which is not available when we are considering the convergence globally.*

The update for $\xi$ in NAG-SC Eq. (13) can be also written as

$$\frac{1}{1-\gamma h}\left(\xi_k + hT_{g_k}\mathsf{L}_{g_k^{-1}}\nabla U(g_k)\right) = \left(\xi_{k-1} + hT_{g_{k-1}}\mathsf{L}_{g_{k-1}^{-1}}\nabla U(g_{k-1})\right) - hT_{g_k}\mathsf{L}_{g_k^{-1}}\nabla U(g_k) \quad (40)$$

This inspires us to define the following modified energy:

$$E^{\text{NAG-SC}}(g,\xi) = U(g) + \frac{(1-\gamma h)^2}{2(1+\gamma h - \gamma^2 h^2)}\left\|\xi + hT_g\mathsf{L}_{g^{-1}}\nabla U(g)\right\|^2 \quad (41)$$

**Theorem 37** (Monotonely decreasing of modified energy of NAG-SC). *Assume the potential $U$ is globally L-smooth. When $h \le \min\left\{\frac{1}{\gamma}, \frac{\gamma}{2L}\right\}$,*

$$E^{NAG\text{-}SC}(g_k, \xi_k) - E^{NAG\text{-}SC}(g_{k-1}, \xi_{k-1}) \le 0$$

*where the modified energy $E^{NAG\text{-}SC}$ is defined in Eq. (41).*

*Proof of Thm. 37.* Given L-smoothness of $U$, we have

$$E^{\text{NAG-SC}}(g_k, \xi_k) - E^{\text{NAG-SC}}(g_{k-1}, \xi_{k-1})$$

$$= U(g_k) - U(g_{k-1}) + \frac{(1-\gamma h)^2}{2(1+\gamma h - \gamma^2 h^2)}\left(\left\|\xi_k + hT_{g_{k-1}}\mathsf{L}_{g_{k-1}^{-1}}\nabla U(g_{k-1})\right\|^2 - \left\|\xi_{k-1} + hT_{g_{k-2}}\mathsf{L}_{g_{k-2}^{-1}}\nabla U(g_{k-2})\right\|^2\right)$$

$$= U(g_k) - U(g_{k-1}) + \frac{(1-\gamma h)^2}{2(1+\gamma h - \gamma^2 h^2)}\left\|\xi_k + hT_{g_{k-1}}\mathsf{L}_{g_{k-1}^{-1}}\nabla U(g_{k-1})\right\|^2$$

$$- \frac{(1-\gamma h)^2}{2(1+\gamma h - \gamma^2 h^2)}\left\|\frac{1}{1-\gamma h}\xi_k + \frac{h(2-\gamma h)}{1-\gamma h}T_{g_{k-1}}\mathsf{L}_{g_{k-1}^{-1}}\nabla U(g_{k-1}))\right\|^2$$

$$= U(g_k) - U(g_{k-1}) + \frac{(1-\gamma h)^2}{2(1+\gamma h - \gamma^2 h^2)}\left\|\xi_k + hT_{g_{k-1}}\mathsf{L}_{g_{k-1}^{-1}}\nabla U(g_{k-1})\right\|^2$$

$$- \frac{1}{2(1+\gamma h - \gamma^2 h^2)}\left\|\xi_k + h(2-\gamma h)T_{g_{k-1}}\mathsf{L}_{g_{k-1}^{-1}}\nabla U(g_{k-1}))\right\|^2$$

$$= U(g_k) - U(g_{k-1}) - \frac{\gamma h(2-\gamma h)}{2(1+\gamma h - \gamma^2 h^2)}\left\|\xi_k\right\|^2 - h\left\langle\xi_k, T_{g_{k-1}}\mathsf{L}_{g_{k-1}^{-1}}\nabla U(g_{k-1})\right\rangle$$

$$- \frac{h^2(3-2\gamma h)}{2(1+\gamma h - \gamma^2 h^2)}\left\|\nabla U(g_{k-1})\right\|^2$$

By L-smoothness, $U(g_k) - U(g_{k-1}) - h\left\langle\xi_k, T_{g_{k-1}}\mathsf{L}_{g_{k-1}^{-1}}\nabla U(g_{k-1})\right\rangle \le \frac{Lh^2}{2}\left\|\xi_k\right\|^2$

$$E^{\text{NAG-SC}}(g_k, \xi_k) - E^{\text{NAG-SC}}(g_{k-1}, \xi_{k-1})$$

$$\le \frac{Lh^2}{2}\left\|\xi_k\right\|^2 - \frac{\gamma h(2-\gamma h)}{2(1+\gamma h - \gamma^2 h^2)}\left\|\xi_k\right\|^2 - \frac{h^2(3-2\gamma h)}{2(1+\gamma h - \gamma^2 h^2)}\left\|\nabla U(g_{k-1})\right\|^2$$

Consequently, a sufficient condition for $E^{\text{NAG-SC}}(g_k, \xi_k) - E^{\text{NAG-SC}}(g_{k-1}, \xi_{k-1}) \le 0$ can be given by

$$\frac{Lh^2}{2} \le \frac{\gamma h(2-\gamma h)}{2(1+\gamma h - \gamma^2 h^2)}$$

By assuming $\gamma h \leq 1$, a sufficient condition for this can be given by $h \leq \frac{\gamma}{2L}$, i.e., when

$$h \leq \min\left\{\frac{1}{\gamma}, \frac{\gamma}{2L}\right\}$$

$\square$

