# OpenReview forum: "Quantitative Convergences of Lie Group Momentum Optimizers"
_NeurIPS.cc/2024/Conference — NeurIPS 2024 poster_

### Official Review · Reviewer_zKf2 · 2024-06-21

**Soundness:** 3
**Presentation:** 2
**Contribution:** 2
**Rating:** 5
**Confidence:** 4

**Summary:**

The authors design and analyze momentum-based algorithms on Lie groups.  They first study ODEs and provide convergence rates for them.  Then they discretize the ODEs (in two different ways -- Lie Heavy Ball and Lie NAG-SC), and show that the second discretization (Lie NAG-SC) has a *locally* accelerated convergence rate.

**Strengths:**

The paper is decently written (but not exceptional).  The most interesting feature of this paper is that one can use the group structure of the Lie group to design optimization algorithms, and avoid parallel transport or the log map usually present in the more general Riemannian setting.

**Weaknesses:**

(1) The requirement that the manifold is a Lie group is very restrictive.  There are many manifolds used in practice which are not Lie groups but still have very nice structure.  Perhaps the prime example is the Stiefel manifolds (which include spheres).  In many applications (especially those related to low rank problems), it is desirable to work with rectangular matrices like Stiefel matrices because storage required for them is much smaller than square matrices (like in SO(n)).  Can the authors provide examples of Lie groups used in practice which are not SO(n), U(n), or products of those?

(2) The paper fails to cite *many* important prior works, most notably (in this order):

* "Accelerated gradient methods for geodesically convex optimization: Tractable algorithms and convergence analysis" by Jungbin Kim and Insoon Yang 2022.  This is the first paper to truly achieve "acceleration" on Riemannian manifolds, specifically having *global* complexity guarantees which scale like O(sqrt{condition number}) or O(sqrt{1/epsilon}).  All prior work only achieved acceleration locally, which is *arguably* not too interesting (see below).

* The updated version of "Global Riemannian Acceleration in Hyperbolic and Spherical Spaces" by Martinez-Rubio also provides global acceleration rates (albeit only on spaces of constant curvature).  [Previous version had exponential dependence on the curvature/radius; this was reduced to a polynomial dependence in the updated version.]

* Lower bounds and obstructions for acceleration on Riemannian manifolds: (a) "No-go Theorem for Acceleration in the Hyperbolic Plane" by Hamilton and Moitra, (b) "Negative curvature obstructs acceleration for strongly geodesically convex optimization, even with exact first-order oracles" by Criscitiello and Boumal, (c) "Curvature and complexity: Better lower bounds for geodesically convex optimization" by Criscitiello and Boumal

* Acceleration in the nonconvex case: "An accelerated first-order method for non-convex optimization on manifolds" by Criscitiello and Boumal

(3) The paper only provides local acceleration, which is *arguably* not too interesting (at least from a query-complexity viewpoint) because locally manifolds look like Euclidean spaces.  This can be made rigorous: there is a generally method for converting Euclidean algorithms to Riemannian ones which locally have the same convergence guarantees: see for example Appendix D of "Curvature and complexity: Better lower bounds for geodesically convex optimization" by Criscitiello and Boumal.

(4) I am not 100% convinced that the Lie Heavy Ball does not have a locally accelerated rate.  My impression is that, with the right choice of parameters, Heavy Ball is *locally* accelerated (essentially because locally a strongly convex cost function looks like a quadratic): see for example "Provable Acceleration of Heavy Ball beyond Quadratics for a Class of Polyak-Łojasiewicz Functions when the Non-Convexity is Averaged-Out" by Wang, Lin, Wibisono, Hu.  On the other hand, the provided numerical experiments seem to indicate that Lie Heavy Ball is not accelerated (but maybe a different choice of parameters would improve convergence?).

(5) For SO(n) or U(n), can the authors please explain why the log map and parallel transport are prohibitively costly (in comparison to the exponential)?

(6) Is it necessary to use the exponential map as the retraction?  From the reviewer's experience, for the manifold SO(n), using the matrix exponential is more expensive than using the QR-retraction.

(7) The experiments performed (the eigen decomposition problem) are limited.  The authors say the eigen decomposition problem "is a hard non-convex problem on a manifold".  The reviewer somewhat disagrees: in some sense, it is one of the easiest nonconvex optimization problems as all second order critical points (where gradient = 0, Hessian is PSD) are global minima.

**Questions:**

Questions and comments (some of which were already asked in the "weaknesses" section):

* Explicitly, what do your algorithms look like when the Lie group is a torus (product of circles)?  Does it reduce to standard NAG?

* Can the authors provide examples of Lie groups used in practice which are not SO(n), U(n), or products of those?

* I am not 100% convinced that the Lie Heavy Ball does not have a locally accelerated rate.  My impression is that, with the right choice of parameters, Heavy Ball is *locally* accelerated (essentially because locally a strongly convex cost function looks like a quadratic): see for example "Provable Acceleration of Heavy Ball beyond Quadratics for a Class of Polyak-Łojasiewicz Functions when the Non-Convexity is Averaged-Out" by Wang, Lin, Wibisono, Hu.  On the other hand, the provided numerical experiments seem to indicate that Lie Heavy Ball is not accelerated (but maybe a different choice of parameters would improve convergence?).

* For SO(n) or U(n), can the authors please explain why the log map and parallel transport are prohibitively costly (in comparison to the exponential)?

* Is it necessary to use the exponential map as the retraction?  From the reviewer's experience, for the manifold SO(n), using the matrix exponential is more expensive than using the QR-retraction.

* Line 277 typo: decreaing -> decreasing

**Limitations:**

No potential negative societal impacts.

---

> ### Author Rebuttal · Authors · 2024-08-07
>
> > W1: The requirement that the manifold is a Lie group is very restrictive. There are many manifolds used in practice which are not Lie groups but still have very nice structure. Perhaps the prime example is the Stiefel manifolds. In many applications (especially those related to low rank problems), it is desirable to work with rectangular matrices like Stiefel matrices because storage required for them is much smaller than square matrices (like in SO(n)).
>
> > Q2: Can the authors provide examples of Lie groups used in practice which are not SO(n), U(n), or products of those?
>
> Thanks for an opportunity of explanation.
>
> 1) *Stiefel versus SO(n).* We agree Stiefel is a very important and useful manifold, but it does not mean SO(n) is not important. Even simple Lie groups like SO(2) and SO(3) are extremely important to applications. For example, machine-learning-based design and generation of molecules (or more precisely, molecular configurations) have already become a big enterprise and a billion-dollar industry. In the majority of research in this direction (e.g., [R5,R6]), side chains are modeled by torsional angles (each in SO(2)), and backbone is modeled by rotational angles (each in SO(3)).
>
> 2) *Lie groups in practice that are not SO(n), U(n), or their products.* Heisenberg group plays an important role in quantum mechanics. Lorentz group plays an important role in general relativity. Spin group is necessary for describing Fermions. Symplectic group is used not only in mechanics but also in quantum computing and even cryptography. Projective linear group can find abundant applications in graphics and visions.
>
> > **W2**: The paper fails to cite many important prior works, most notably (in this order):
>
> We appreciate all the references and we will cite them in a revision. However, our paper focuses on a different algorithm under a different problem setting, and the existence of these paper does not weaken our novelty. Kindly see details below:
>
> > "Accelerated gradient methods for geodesically convex optimization: Tractable algorithms and convergence analysis" by Jungbin Kim and Insoon Yang 2022. This is the first paper to truly achieve "acceleration" on Riemannian manifolds, specifically having global complexity guarantees which scale like O(sqrt{condition number}) or O(sqrt{1/epsilon}). All prior work only achieved acceleration locally, which is arguably not too interesting (see below).
>
> This is an interesting paper, however, the proposed algorithm (Algo 1 & 2 in their paper) requires logarithm (which may not be uniquely defined globally on compact manifolds) and the expensive parallel transport. What's more, its global acceleration requires global convexity, which is too strong in our case.
>
> > The updated version of "Global Riemannian Acceleration in Hyperbolic and Spherical Spaces" by Martinez-Rubio also provides global acceleration rates (albeit only on spaces of constant curvature). [Previous version had exponential dependence on the curvature/radius; this was reduced to a polynomial dependence in the updated version.]
>
> Their algorithm (Algo. 1) requires a line search in step 6, which can be hard when applied to complex problems (large-scale and non-convex). In contrast, our algorithm can be easily applied to machine learning problems, e.g., vision transformers in general reply to all.
>
> > Lower bounds and obstructions for acceleration on Riemannian manifolds: (a) "No-go Theorem for Acceleration in the Hyperbolic Plane" by Hamilton and Moitra, (b) "Negative curvature obstructs acceleration for strongly geodesically convex optimization, even with exact first-order oracles" by Criscitiello and Boumal, ( c) "Curvature and complexity: Better lower bounds for geodesically convex optimization" by Criscitiello and Boumal
>
> These interesting papers will be cited. In fact, these negative results further illustrate the nontriviality of our result, which is a positive one, and there is no contradiction due to different setups.
>
> > Acceleration in the nonconvex case: "An accelerated first-order method for non-convex optimization on manifolds" by Criscitiello and Boumal
>
> This paper is interesting because it uses the idea of optimizing on the tangent space and utilizing the accelerated algorithms in Euclidean spaces. However, due to this reason, many of its conditions (e.g., A2 on page 14) depend on the local chart and can be hard to check. Instead, for our algorithm, the condition of the objective function, the choice of hyperparameters and the convergence rate are all intrinsic.
>
> > **W3**: The paper only provides local acceleration, which is arguably not too interesting (at least from a query-complexity viewpoint) because locally manifolds look like Euclidean spaces. This can be made rigorous: there is a generally method for converting Euclidean algorithms to Riemannian ones which locally have the same convergence guarantees: see for example Appendix D of "Curvature and complexity: Better lower bounds for geodesically convex optimization" by Criscitiello and Boumal.
>
> Obtaining a uniform, nonasymptotic bound for the global convergence for the optimization of general nonconvex functions, as far as we know, is still an open question even in Euclidean space. One often needs to work with specific objective function(s) and even consider special initalization (e.g., [R4]) in order to get accurate rates; note high accuracy bound is needed as we are investigating whether there is acceleration in this paper. We agree that for convex problems, such as studied in the nice paper `Accelerated gradient methods for geodesically convex optimization: Tractable algorithms and convergence analysis" by Jungbin Kim and Insoon Yang 2022`, it is possible to get global result (which is remarkable), but the setup considered in this paper is very different, as we consider accelerated optimization of general **non**convex functions in **curved** space. We are not aware of any result that can give global rate in this case.

---

> ### Author Response · Authors · 2024-08-07
> **Rebuttal to zKf2 by Authors (Part 2/2)**
>
> > **W4, Q3**: I am not 100% convinced that the Lie Heavy Ball does not have a locally accelerated rate. My impression is that, with the right choice of parameters, Heavy Ball is locally accelerated: see for example [Wang, Lin, Wibisono, Hu]. On the other hand, the provided numerical experiments seem to indicate that Lie Heavy Ball is not accelerated (but maybe a different choice of parameters would improve convergence?).
>
> We are sorry for the confusion, our acceleration means acceleration under strong convexity assumption. [Wang, Lin, Wibisono, Hu], although establish a similar convergence rate $1-1/\sqrt{\kappa}$, focus on different assumptions, some of them are strong, e.g, diagonal Hessian in Thm. 1 in their paper. In fact, they state in that paper that HB fails to accelerate under strong convex assumption by 'In convex optimization, it is known that Nesterov’s momentum and HB share the same ODE in continuous time (Shi et al., 2021). Yet, the acceleration disappears when one discretizes the dynamic of HB and bounds the discretization error.'
>
> In fact, for the heavy-ball method, although there exist some other choices of hyperparameters, leading to acceleration on quadratic functions, they cannot be applied to a general strongly convex function. A summary can be found in [R7].
>
> > **W5, Q4**: For SO(n) or U(n), can the authors please explain why the log map and parallel transport are prohibitively costly (in comparison to the exponential)?
>
> Sorry for the confusion. They are costly because they are **additional** operations that are avoided by our method. All accelerated 1st-order manifold optimizers that we're aware of use at least one exponential map per iteration, and that is the main total cost of our algorithm. However, all other operations, i.e., logarithms, parallel transport, extra exponentials, can be understood as 'expensive', even if their computational complexity is not worse. An empirical estimation of the cost is, the NAG-SC in [R1] costs around 5 times more time than our algorithm.
>
> In addition, logarithm may cause some trouble when implementing the algorithm due to the lack of unique geodesic on most Lie groups. Regarding parallel transport, similiar to the general manifold case, the parallel transport on Lie groups is also defined by an ODE and requires costly numerical integration (an expression for parallel transport on Lie groups with left-invariant metric can be found in Thm. 1 in [R8]).
>
>
> > **W6, Q5**: Is it necessary to use the exponential map as the retraction? From the reviewer's experience, for the manifold SO(n), using the matrix exponential is more expensive than using the QR-retraction.
>
> We appreciate this suggestion for using cheaper retractions, e.g., Cayley map and QR retraction. However, even though we may be able to provide some of the empirical results, we are unsure if theoretical results will be optimistic. This is because acceleration is fragile: NAG-SC and HB are only different in an $O(h^2)$ term and lead to a totally different dependence on condition number. However, retractions will introduce numerical errors whose effect is unknown and hard to analyze.
>
> > **W7**: The experiments performed (the eigen decomposition problem) are limited. The authors say the eigen decomposition problem "is a hard non-convex problem on a manifold". The reviewer somewhat disagrees: in some sense, it is one of the easiest nonconvex optimization problems as all second order critical points (where gradient = 0, Hessian is PSD) are global minima.
>
> Thank you for pointing out this and we will correct this in the next version. We admit eigen decomposition is not a good example for avoiding local minimum, however, we are mostly focusing on the rate of convergence and we will add discussions about all local minimum are global minimum in the parts focusing on global convergence. We have additional numerical experiment including other algorithms and more complicated problems. Please see the general rebuttal to all.
>
> > **Q1**: Explicitly, what do your algorithms look like when the Lie group is a torus (product of circles)? Does it reduce to standard NAG?
>
> Yes, it will degenerate to Euclidean NAG-SC if d-torus is unrolled into and represented as $\mathbb{R}^d$.
>
> > **Q6**: Line 277 typo: decreaing -> decreasing
>
> We appreciate the careful review and are sorry for the embarrassing typo.
>
> [R4] Ward and Kolda. Convergence of alternating gradient descent for matrix factorization. NeurIPS'23
>
> [R5] Watson et al. De novo design of protein structure and function with RFdiffusion. Nature 2023
>
> [R6] Stark et al. Harmonic Self-Conditioned Flow Matching for Multi-Ligand Docking and Binding Site Design. ICML 2024
>
> [R7] Lessard, Laurent, Benjamin Recht, and Andrew Packard. "Analysis and design of optimization algorithms via integral quadratic constraints."
>
> [R8] Nicolas Guigui and Xavier Pennec. A reduced parallel transport equation on lie groups with a left-invariant metric

---

> > ### Comment · Reviewer_zKf2 · 2024-08-10
> > **Reviewer response to authors (1)**
> >
> > I appreciate the authors’ detailed response.  However, given the overall contribution and novelty of this paper, I will maintain my score.  Comments:
> > * The authors satisfactorily addressed several of my questions, for example, explaining why the (Euclidean or Lie) Heavy Ball does not have a locally accelerated rate.
> > * However, other concerns remain: for example, the requirement that the manifold is a Lie group is very restrictive (and hard to avoid given the paper's scope).

---

> > > ### Author Response · Authors · 2024-08-12
> > >
> > > > The authors satisfactorily addressed several of my questions, for example, explaining why the (Euclidean or Lie) Heavy Ball does not have a locally accelerated rate.
> > >
> > > We sincerely thank the reviewer for discussing with us and acknowledging the validity of our statements.
> > >
> > > > However, other concerns remain: for example, the requirement that the manifold is a Lie group is very restrictive (and hard to avoid given the paper's scope).
> > >
> > > We completely agree that this paper is specifically about Lie group, not general manifold. However, we very much hope the reviewer could agree with us that it is about **quantitative**, **accelerated**, **nonconvex** optimization. Each word alone already carries a lot of weight and has been traditionally well appreciated by the machine learning theory community; for example, [R10] (published in NeurIPS'18) focuses on acceleration under convex functions of Runge-Kutta discretization; [R11] (published in COLT'18) focuses on local acceleration on curved spaces; [R12] (published in ICLR'19) focuses on designing adaptive learning rate method with provable convergence under global convexity. More importantly, at this moment, we're unaware of any result that can do them all in once for general manifolds. It is widely accepted by the community that focusing on a subclass of problems that have more structure is still insightful; for example, the nice reference [R9] suggested by the reviewer him/herself focuses on (no) acceleration in optimization on a very specific case (globally convex functions, and 2-dim hyperbolic plane **only**), but it is a great work in our opinion.
> > >
> > > Moreover, for general Riemannian manifold optimization, one can (and typically do) assume geodesic convexity or its relaxation such as convexity outside a ball, but these assumptions cannot be made for compact Lie groups we considered (See Rmk. 1 on page 3 in our original submission). Therefore, restricting to Lie group is not only making the problem easier, but also making it harder at the same time.
> > >
> > > But perhaps the reviewer's remaining concern mainly lies in why is Lie group optimization important. Here are some answers:
> > > 1) There are already a lot of important machine learning applications in the literature. For example, [R13] demonstrates imposing SO(n) constraint on weight matrices are beneficial for deep networks; [R14] proves orthogonality benefits deep CNNs. [R15] shows artificial orthogonal constraints in RNN improve long-term dependencies. [R16] finds that rotation activation or weight matrices help remove outliers and benefit quantization on large language models; [R17] shows artificial orthogonal constraints improve robustness. All these amount to optimization on Lie groups.
> > > 2) In the rebuttal supplementary pdf, we already provided an additional application of Lie group optimzation. It show-cased how high-dimensional accelerated Lie group optimization can significantly improve vanilla attention mechanism, e.g., Lie group optimization boosts the performance of ViT: CIFAR 10 error improves from 9.75% (by Euclidean optimization; see [R2]) to 8.89% (by our proposed Lie NAG-SC)/ 9.46% (by Lie HB that has no acceleration) and CIFAR 100 error improves from 32.61% (by Euclidean optimization; see [R2]) to  31.11% (by our proposed Lie NAG-SC)/ 31.72% (by Lie HB that has no acceleration). Numbers for Lie HB and Lie NAG-SC are from the rebuttal supplementary pdf.
> > > 3) We also provided additional examples in the first round of rebuttal [R5, R6] that demonstrate how Lie group generative modeling is creating a big industry, but we now realize that we underexplained (apology) the connection between (Lie group) generative modeling and optimization, so please allow us to do it again: very briefly put, to use diffusion model for generative modeling on Lie group to enable important applications, one needs to have a forward dynamics that can push data distribution forward to an easy-to-sample distribution, and one can do so by first finding an (in this case, a good Lie group) optimizer with momentum, and then adding noise to it to turn it into a sampler, and finally using this sampler as the forward dynamics of a diffusion generative model; see [R18] for details.
> > >
> > > We truly hope this can clarify why we plea the reviewer to kindly reconsider our overall contribution and novelty.
> > >
> > > (Please see the references in our next comment due to the character limit)

---

> > > > ### Author Response · Authors · 2024-08-12
> > > > **References in the last comment**
> > > >
> > > > We appreciate the reviewer's effort in helping improve our paper. Please find the references in our last comment in the following:
> > > >
> > > > [R2] Lingkai Kong, Yuqing Wang, and Molei Tao. "Momentum stiefel optimizer, with applications to suitably-orthogonal attention, and optimal transport." ICLR 2023.
> > > >
> > > > [R5] Watson et al. De novo design of protein structure and function with RFdiffusion. Nature 2023
> > > >
> > > > [R6] Stark et al. Harmonic Self-Conditioned Flow Matching for Multi-Ligand Docking and Binding Site Design. ICML 2024
> > > >
> > > > [R9]Linus Hamilton and Ankur Moitra. "A no-go theorem for robust acceleration in the hyperbolic plane." NeurIPS, 2021.
> > > >
> > > > [R10] Jingzhao Zhang et al. "Direct Runge-Kutta discretization achieves acceleration." NeurIPS 2018.
> > > >
> > > > [R11] Hongyi Zhang and Suvrit Sra. "An estimate sequence for geodesically convex optimization." COLT, 2018.
> > > >
> > > > [R12] Gary Becigneul and Octavian-Eugen Ganea. "Riemannian Adaptive Optimization Methods." ICLR, 2019.
> > > > [R13] A. Saxe, J. McClelland, and S. Ganguli. "Exact solutions to the nonlinear dynamics of learning in deep linear neural networks." ICLR 2014.
> > > >
> > > > [R14] Lechao Xiao et al. "Dynamical isometry and a mean field theory of cnns: How to train 10,000-layer vanilla convolutional neural networks." ICML, 2018.
> > > >
> > > > [R15] Mario Lezcano-Casado and David Martınez-Rubio. "Cheap orthogonal constraints in neural networks: A simple parametrization of the orthogonal and unitary group." ICML, 2019.
> > > >
> > > > [R16] Zechun Liu et al. "SpinQuant--LLM quantization with learned rotations." arXiv preprint arXiv:2405.16406
> > > >
> > > > [R17] Moustapha Cisse et al. "Parseval networks: Improving robustness to adversarial examples." ICML, 2017.
> > > >
> > > > [R18] Zhu et al. Trivialized Momentum Facilitates Diffusion Generative Modeling on Lie Groups. arXiv: 2405.16381

---

### Official Review · Reviewer_JEvs · 2024-06-26

**Soundness:** 3
**Presentation:** 3
**Contribution:** 3
**Rating:** 7
**Confidence:** 3

**Summary:**

This work first analyzes the convergence rate of the Lie group momentum optimizer by applying the techniques from optimization theory over manifolds to optimization over Lie groups and extending the Lyapunov analysis to Lie group settings. They also provide the convergence analysis of the discrete version of the above dynamics by constructing new energy function and a new Lyapunov function. Besides, they extend an acceleration technique from Euclidean case to Lie group case and prove its performance.

**Strengths:**

* The authors apply the theories from optimization over general manifolds to considering optimization over Lie groups and carefully analyze the structure of Lie groups such that the general theories have more analytical and tractable formulas. Moreover, they extend techniques from the Euclidean case, like the Lyapunov analysis, and Heavy-Ball algorithm, to the Lie group case, which is both intuitive and rigorous.

* The constructions of the discrete version of the energy function and the Lyapunov function from the continuous version of these functions are insightful and it may be helpful for us to consider a similar problem related to a discrete dynamic system.

* The authors provide intuitive explanations for almost every theory, from which I can see the motivations and it is helpful for understanding the whole picture behind the technical details.

**Weaknesses:**

* In line $280$ and the proof in the appendix, the authors mentioned the ''curvature'' but the article does not provide much information about that. I would like to know if this ''curvature'' is an intuitive term related to the second-order information or a rigorous term related the curvature information of the Riemannian structure of $\mathtt{G}$.

**Questions:**

* In the article, a function $U$ is convex means it is convex in the meaning of Euclidean case, even it is defined on Lie group $\mathtt{G}$. The convexity means $U$ is convex on $\mathtt{G}$, where $\mathtt{G} \subset \mathbb{R}^N$ is embedded in Euclidean space. Is that right?

* For equation $(16)$, does $d_{\xi} \log g$ means $(d \log)_{g}(\xi)$, i.e. the differential of the logarithm at point $g$ mapping $\xi$ ?

* There may be a typo in equation $(18)$. Is the following statement right? $A := \max_{\left\lVert X \right\lVert = 1} \sigma(\operatorname{ad}_X)$, where $\sigma(\cdot)$ is set of all eigenvalues.

* About the **Assumption 2**, does Lie group $\mathtt{G}$ need to be connected?

**Limitations:**

The authors adequately addressed the limitations.

---

> ### Author Rebuttal · Authors · 2024-08-07
>
> > **W1**: In line 280 and the proof in the appendix, the authors mentioned the ''curvature'' but the article does not provide much information about that. I would like to know if this ''curvature'' is an intuitive term related to the second-order information or a rigorous term related the curvature information of the Riemannian structure of $G$.
>
> It is a rigorous term related to the curvature information of the Riemannian structure of $G$. $p(a)$ is explicitly defined in Eq. 17, and $a$ is defined in line 272 in Thm 14, whose definition depends on $A$ (defined in Eq. 18). Under the inner product making $\operatorname{ad}$ skew-adjoint in Lemma 3, $A=\sqrt{\text{max sectional curvature}}/4$  (Please see Sec. D.2 in [R3] in the reference for more details.)
>
>
> > **Q1**: In the article, a function $U$ is convex means it is convex in the meaning of Euclidean case, even it is defined on Lie group $G$. The convexity means $U$ is convex on $G$, where $G\subset\mathbb{R}^n$ is embedded in Euclidean space. Is that right?
>
> The short answer is no. The convexity under discussion is geodesic convexity, which is different from convexity in ambient Euclidean space after embedding. To illustrate the drastic difference, for example, convex functions on Euclidean spaces must be discussed on a convex set, however, we can still have convex functions on a manifold even if its Euclidean embedding is non-convex.
>
> > **Q2**: For equation (16), does $d_\xi\log g$ means $(d\log)_g(\xi)$, i.e. the differential of the logarithm at point mapping ?
>
> Yes, the expert reviewer is totally correct.
>
> > **Q3**: There may be a typo in equation (18). Is the following statement right? $A:=\max_{\|x\|=1}\sigma(\operatorname{ad}_X)$, where $\sigma(\cdot)$ is set of all eigenvalues.
>
> We appreciate the careful review and will correct the embarrassing typo.
>
> > **Q4**: About the Assumption 2, does Lie group $G$ need to be connected?
>
> Thanks for a great question. We understand the rationale behind this question, but we don't actually require connectness. The reason is: to prove global convergence, we prove and leverage the monotonicity of an energy function, whose definition does not require connectness. For convergence rate, it is only quantified asymptotically and it does not make a difference with or without connectness.
>
> [R3] Lingkai Kong and Molei Tao. Convergence of kinetic langevin monte carlo on lie groups. COLT, 2024.

---

> > ### Comment · Reviewer_JEvs · 2024-08-11
> >
> > Thank the authors for clearifying my concerns about the curvature information. And it is better to include this contents in the appendix and provide more information.

---

> > > ### Author Response · Authors · 2024-08-11
> > > **Thanks for "Official Comment by Reviewer JEvs"**
> > >
> > > We thank the expert again for recognizing our contributions and helping us further improve the quality of our paper.
> > >
> > > Yes, we will certainly include this content in the appendix with more information.

---

### Official Review · Reviewer_dors · 2024-07-11

**Soundness:** 2
**Presentation:** 1
**Contribution:** 1
**Rating:** 1
**Confidence:** 5

**Summary:**

The authors analyze the momentum method and Nesterov accelerated gradient descent method on the Lie group. With some knowledge of Riemannian geometry, they discuss the computational cost.

**Strengths:**

I have found none strenghs.

**Weaknesses:**

I do not know the author's motivation to write this manuscript.

Could you show me a reasonable example for application in practice, or close to practice?

There are so many high-level mathematical terminologies, but no essential problems are solved.

**Questions:**

I think this manuscript is none sense, so I do not ask any questions.

**Limitations:**

The content are none sense.

---

### Official Review · Reviewer_K3KR · 2024-07-12

**Soundness:** 3
**Presentation:** 3
**Contribution:** 3
**Rating:** 6
**Confidence:** 4

**Summary:**

This paper explores the optimization of functions defined on Lie groups using momentum-based dynamics. The authors propose two discretization methods, Lie Heavy-Ball and Lie NAG-SC, and analyze their convergence rates.

The main contributions are as follows:
1. Provide the first quantitative analysis of Lie group momentum optimizers.
2. Theoretically show an intuitively constructed momentum optimizer, namely Lie Heavy-Ball, may not yield accelerated convergence.
3. Generalize technique from Euclidean optimization to propose a Lie group optimizer that provably has acceleration.

**Strengths:**

1. The authors provide the first quantitative analysis of Lie group momentum optimizers which is significant, since there is no nontrivial convex functions on many Lie groups.
2. Theoretically show an intuitively constructed momentum optimizer, namely Lie Heavy-Ball, may not yield accelerated convergence.
3. Generalize technique from Euclidean optimization to propose a Lie group optimizer, Lie NAG-SC, that provably has acceleration.
4. Comparing to other optimizers that are designed for general manifolds, the proposed approach bypasses the requirements for costly operations.

**Weaknesses:**

1. The idea of the paper is natural. I think it can be seen as a straightforward extension of the results in [1]. So it may be not novelty.
2. More emperical results are needed I think. Also it better show the comparision with the result in [1].





[1]Tao, Molei, and Tomoki Ohsawa. "Variational optimization on lie groups, with examples of leading (generalized) eigenvalue problems." International Conference on Artificial Intelligence and Statistics. PMLR, 2020.

**Questions:**

As I mentioned in the weaknesses, the motivation should be more clear. I want to know the novelty of this work compared with [1]

Is it possible show more experiment results in other problems?

Comparision with the experiment results in [1]?

[1]Tao, Molei, and Tomoki Ohsawa. "Variational optimization on lie groups, with examples of leading (generalized) eigenvalue problems." International Conference on Artificial Intelligence and Statistics. PMLR, 2020.

**Limitations:**

See the weaknesses.

---

> ### Author Rebuttal · Authors · 2024-08-07
>
> > W1: The idea of the paper is natural. I think it can be seen as a straightforward extension of the results in [Tao & Ohsawa]. So it may be not novelty.
>
> > Q1: I want to know the novelty of this work compared with [Tao & Ohsawa]
>
> [Tao & Ohsawa] is definitely an inspiration to this work. However, their algorithm is essentially the same as our Lie Heavy-Ball (Rmk. 28) and may not have accelerated convergence. The novelties of our paper are
> 1) quantification of the convergence rate of Lie Heavy-Ball (not enough acceleration),
> 2) a solution, which is a new algorithm Lie NAG-SC,
> 3) quantification of the convergence rate of the new algorithm (true acceleration).
>
> In addition, there is a strong technical contribution, namely all the convergence analyses have to be done for fully non-convex functions, because there is in general no non-constant convex functions on compact Lie groups. Rigorous analysis of nonconvex optimization is always challenging, let alone this time on manifold as well.
>
>
> > W2. More emperical results are needed I think. Also it better show the comparision with the result in [Tao & Ohsawa].
>
> > Q2. Is it possible show more experiment results in other problems?
>
> Thank you for the suggestion and we perform more numerical experiments. Please kindly refer to the general rebuttal to all and the rebuttal pdf supplement.
>
> > Q3: Comparision with the experiment results in [Tao & Ohsawa]?
>
> Thank you for this suggestion. Lie Heavy-Ball mentioned in our paper is essentially the algorithm in [Tao & Ohsawa] (we will clarify this in a revision). In our section 6.2, [Tao & Ohsawa]/Lie Heavy-Ball is already experimentally compared with our newly proposed Lie NAG-SC.
>
> The general rebuttal to all and Fig. 2 in the the rebuttal pdf supplement contain more experiments, where Lie HB and Lie NAG-SC are applied to (and compared on) vision transformers on Cifar dataset.

---

> > ### Comment · Reviewer_K3KR · 2024-08-10
> >
> > Thank you for your clarifications and additional materials. It makes me understand it clearly.

---

> > > ### Author Response · Authors · 2024-08-10
> > >
> > > Thank you for your valuable time and kind consideration!

---

### Official Review · Reviewer_m3jJ · 2024-07-13

**Soundness:** 3
**Presentation:** 3
**Contribution:** 3
**Rating:** 7
**Confidence:** 3

**Summary:**

The paper proposes a new algorithm (Lie NAG-SC) that converges at accelerated rates on Lie groups when initiated close to the true optimum (local convergence). Theoretical analysis and experimental verification shows good performance.

**Strengths:**

The theoretical analysis is quite advanced, and as far as we are aware of, this is the first method to achieve local accelerated rates for Lie groups.

**Weaknesses:**

Both the theory and the experimental evaluation only considered local convergence, i.e. cases when the optimizer is initiated near the global optimum. This is a limitation as the relative behaviour of the algorithms further away from the optimum may be completely different.

The impact of the parameter p(a) on the rates is not sufficiently clearly discussed. This is related to the curvature of the manifold, as well as how closely it is initiated to the true optimum.

**Questions:**

Would you be able to numerically compare the different methods when started from the same random initialisation (i.e. same initial position), far away from the global optimum?

Could you discuss the impact of the parameter p(a) on the rates, and explain the intuition for this parameter in the introduction?
You should not claim acceleration without addressing the reduction in rates due to this parameter.

**Limitations:**

Limitations have been clearly addressed.

---

> ### Author Rebuttal · Authors · 2024-08-07
>
> > **W1**: Both the theory and the experimental evaluation only considered local convergence, i.e. cases when the optimizer is initiated near the global optimum. This is a limitation as the relative behaviour of the algorithms further away from the optimum may be completely different.
>
> We agree the expert's opinion that global convergence is interesting. However, due to the lack of global convexity, it is hard to have a global convergence rate. Even in Euclidean space, as far as we know, obtaining a uniform, nonasymptotic bound for the global convergence for the optimization of general nonconvex functions is still an open question.
>
> > **W2**: The impact of the parameter p(a) on the rates is not sufficiently clearly discussed. This is related to the curvature of the manifold, as well as how closely it is initiated to the true optimum.
>
> > **Q2**: Could you discuss the impact of the parameter p(a) on the rates, and explain the intuition for this parameter in the introduction? You should not claim acceleration without addressing the reduction in rates due to this parameter.
>
>
> As shown in Table 1, $p(a)$ only shows up in the convergence analysis of NAG-SC. The details are in line 280 to 284. In short, $p(a)$ quantifies the loss of convergence induced by the curved space. When close to the minimum, the space is approximately flat. The negative effect of $p(a)$ disappears when $2p(a)<\sqrt{2L}$ (Table 1 and Thm. 14) and the convergence rate will be the same as Euclidean space. This is the reason we claim acceleration. Similar curvature-dependence of convergence rate can also be found in [1] and [27] in the paper. See also Rmk. 35.
>
> > **Q1**: Would you be able to numerically compare the different methods when started from the same random initialisation (i.e. same initial position), far away from the global optimum?
>
> Thank you for the advice and we agree that more numerical results would be helpful. Please see the general rebuttal to all. We added optimizer from Riemannian optimization into comparison and also applied our optimizer to vision transformers with Cifar dataset.

---

> > ### Comment · Reviewer_m3jJ · 2024-08-11
> > **Acknowledgement of rebuttal**
> >
> > We thank the authors for addressing our comments.
> >
> > The new experiments are much more convincing about the usefulness of the algorithm, please include them in the final version of the paper. I am increasing my score to 7.

---

> > > ### Author Response · Authors · 2024-08-11
> > > **Thank you for "Acknowledgement of rebuttal"**
> > >
> > > We sincerely thank the expert reviewer for helping us greatly improve the quality of our paper and your time!

---

### Author Rebuttal · Authors · 2024-08-07

**General rebuttal to all**

> **Q1**: More experimental results

A1: We perform more numerical experiments in the rebuttal PDF supplement.
- We add Riemannian GD and Riemannian NAG-SC [R1] into comparison (Fig. 1 in the attached pdf). Of course, our proposed method converges faster than Riemannian GD which has no momentum nor acceleration. But we also see even faster convergence than Riemannian NAG-SC, which should already have acceleration (however with higher computational cost per step), possibly because our method is specially designed for Lie groups.
- We apply (and compare) the newly proposed Lie NAG-SC method and the existing Lie-Heavy Ball method to a practical machine learning application, namely improving Transformer by requiring attention heads to be orthogonal so that attentions are not redundant [R2].


The training of this modified Transformer amounts to a Lie group optimization problem. We used Lie NAG-SC and Lie HB to train the orthogonal parameters and momentum SGD for the unconstraint parameters. We trained vanilla Vision Transformer from scratch on Cifar till a fixed amount of epochs, and observed improved performance in terms of validation error when Lie HB is replaced by the accelerated method Lie NAG-SC (Cifar 10: 9.46% $\to$ 9.89%, Cifar 100: 31.72% $\to$ 31.11%).

[R1] Kwangjun Ahn and Suvrit Sra. From nesterov’s estimate sequence to riemannian acceleration. In Conference on Learning Theory, pages 84–118. PMLR, 2020.

[R2] Lingkai Kong, Yuqing Wang, and Molei Tao. Momentum stiefel optimizer, with applications to suitably-orthogonal attention, and optimal transport. ICLR, 2023.

---

### Decision · Program_Chairs · 2024-09-25

**Decision:**

Accept (poster)

**Comment:**

From the valid reviews, the reviewers reach a consensus that the paper contains interesting result and novel contribution to the community. The Author-Reviewer discussions were substantial, the authors should incorporate the important ones to the final version of the paper and supplementary. Below are some points (but not limited to) we would like to see
 - Since Lie group is quite restrict, the difficulties and motivation should be better presented. Motivating examples from machine learning related areas would be great.
 - Only local acceleration guarantee, which is understandable due to the nature of the manifold. On the other hand, it seems that practically how to enter the local region is problematic or not clearly discussed in the paper: 1) the locality condition requires $g_*$ which is not realistic; 2) suppose we can solve the first problem via some estimation of $g_*$, how to enter this locally region. For a given problem without prior knowledge, it seems hard.